# Two-way Dispatched function in Sonic hedgehog shedding and transfer to high-density lipoproteins

**Kristina Ehring†, Sophia Friederike Ehlers†, Jurij Froese, Fabian Gude, Janna Puschmann, Kay Grobe***

Institute of Physiological Chemistry and Pathobiochemistry, University of Münster, Münster, Germany

**\*For correspondence:**
kgrobe@uni-muenster.de

†These authors contributed equally to this work

**Competing interest:** The authors declare that no competing interests exist.

**Abstract** The Sonic hedgehog (Shh) signaling pathway controls embryonic development and tissue homeostasis after birth. This requires regulated solubilization of dual-lipidated, firmly plasma membrane-associated Shh precursors from producing cells. Although it is firmly established that the resistance-nodulation-division transporter Dispatched (Disp) drives this process, it is less clear how lipidated Shh solubilization from the plasma membrane is achieved. We have previously shown that Disp promotes proteolytic solubilization of Shh from its lipidated terminal peptide anchors. This process, termed shedding, converts tightly membrane-associated hydrophobic Shh precursors into delipidated soluble proteins. We show here that Disp-mediated Shh shedding is modulated by a serum factor that we identify as high-density lipoprotein (HDL). In addition to serving as a soluble sink for free membrane cholesterol, HDLs also accept the cholesterol-modified Shh peptide from Disp. The cholesteroylated Shh peptide is necessary and sufficient for Disp-mediated transfer because artificially cholesteroylated mCherry associates with HDL in a Disp-dependent manner, whereas an N-palmitoylated Shh variant lacking C-cholesterol does not. Disp-mediated Shh transfer to HDL is completed by proteolytic processing of the palmitoylated N-terminal membrane anchor. In contrast to dual-processed soluble Shh with moderate bioactivity, HDL-associated N-processed Shh is highly bioactive. We propose that the purpose of generating different soluble forms of Shh from the dual-lipidated precursor is to tune cellular responses in a tissue-type and time-specific manner.

## eLife assessment

This **useful** article presents an analysis of different factors that are required for release of the lipid-linked morphogen Shh from cellular membranes. The evidence is still **incomplete**, as experiments rely on over-expression of Shh in a single cell line and are sometimes of a correlative nature. The study, which otherwise confirms and extends previous findings, will be of interest to developmental biologists who work on Hedgehog signaling.

## Introduction

Hedgehog (Hh) ligands activate an evolutionarily conserved signaling pathway that provides instructional cues during tissue morphogenesis and, when misregulated, contributes to developmental disorders and cancer. Hhs are unique in that they require autocatalytic covalent cholesteroylation of the C-terminus (*Porter et al., 1996b*) and N-terminal palmitoylation by a separate Hh acyltransferase (Hhat) activity (*Pepinsky et al., 1998*). Both lipids tightly bind Hh to the plasma membrane of producing cells to effectively prevent unregulated ligand release (*Peters et al., 2004*). Signaling to distant target cells therefore requires the regulated solubilization of Hh from the membrane of

ligand-producing cells, a process that is facilitated by vertebrate and invertebrate Dispatched (Disp) orthologs in vitro (*Ehring et al., 2022*) and in vivo (*Burke et al., 1999*; *Ma et al., 2002*; *Kawakami et al., 2002*; *Nakano et al., 2004*). Disp contains 12 transmembrane helices and 2 extracellular domains and belongs to the resistance-nodulation-division family of transmembrane efflux pumps. Normally, such pumps maintain cellular homeostasis and remove toxic compounds. In addition, Disp contains a conserved multi-pass transmembrane domain known as the sterol sensing domain (SSD), which regulates the homeostasis of free or esterified cellular cholesterol in other SSD proteins (*Hall et al., 2019*). This molecular architecture is consistent with Disp extraction of free cholesterol from the plasma membrane to remove it from the cell (*Ehring et al., 2022*). In addition, given its established role in Hh release, it has been suggested that Disp also extracts the C-terminally linked Hh sterol to solubilize Hh in the extracellular compartment (*Tukachinsky et al., 2012*).

However, Disp activity alone is not sufficient to solubilize the vertebrate Hh family member Sonic hedgehog (Shh) from the plasma membrane. A second synergistic factor required for maximal Shh signaling is the soluble extracellular glycoprotein Scube2 (Signal sequence, cubulin [CUB] domain, epidermal growth factor [EGF]-like protein 2) (*Hall et al., 2019*). One way to explain the Disp/Scube2 synergy is that Disp-mediated extraction of dual-lipidated Shh from the plasma membrane delivers it to Scube2, a mechanism that depends on the C-terminal cysteine-rich and CUB domains of Scube2 (*Tukachinsky et al., 2012*; *Creanga et al., 2012*). In vitro support for this mechanism comes from co-immunoprecipitation of Disp and Scube2 with Shh (*Tukachinsky et al., 2012*; *Creanga et al., 2012*) and from structural data (*Wierbowski et al., 2020*). Other proposed carriers for Disp-extracted lipidated Hh/Shh include lipoprotein particles (LPPs) (*Eugster et al., 2007*; *Palm et al., 2013*; *Panáková et al., 2005*), exosomes as carriers of internalized and re-secreted Hhs (*Gradilla et al., 2014*; *Vyas et al., 2014*; *Matusek et al., 2014*), and micellous Hh assemblies (*Zeng et al., 2001*). Another proposed mode of Shh release is Disp-regulated proteolytic processing (termed shedding) from the plasma membrane (*Ehring et al., 2022*) by two major plasma membrane-associated sheddases, A Disintegrin and Metalloproteinase (ADAM) 10 and 17 (*Dierker et al., 2009*; *Ohlig et al., 2011*; *Damhofer et al., 2015*). The role of Scube2 in the shedding model is to enhance proteolytic processing of both terminal lipidated Shh peptides in a CUB domain-dependent manner (*Jakobs et al., 2014*; *Jakobs et al., 2016*; *Jakobs et al., 2017*). Consistent with this, CUB function in regulated shedding of other substrates often involves substrate recognition and induced structural changes in the substrate to increase turnover (*Bourhis et al., 2013*; *Jakobs et al., 2017*; *Kim et al., 2021*).

In this study, we systematically characterized the biochemical parameters of Disp-regulated Shh solubilization with all these published models in mind. To this end, we used a unique bicistronic Hhat/Shh coexpression system to ensure that only dual-lipidated Shh was produced and analyzed as a substrate for Disp (*Jakobs et al., 2014*). We also avoided any protein tagging to ensure undisturbed Shh modification, secretion, and interaction with Disp and other potential pathway components. Finally, we used unbiased biochemical methods to analyze Shh solubilization, size, and lipidation status under different experimental conditions. First, we confirmed by SDS-PAGE/immunoblotting and reverse-phase high-performance liquid chromatography (RP-HPLC) that Disp and Scube2 synergistically enhance Shh shedding into the cell culture medium (*Jakobs et al., 2014*). We also found that repeated washing of Disp- and Shh-expressing cells to remove all traces of serum abolished Shh release, and that high levels of serum promoted the solubilization of a previously unknown, highly bioactive Shh variant that lacks its palmitoylated N-terminus but retains its cholesteroylated C-terminus. Size-exclusion chromatography (SEC) analyses revealed co-elution of this novel Shh variant with serum apolipoprotein A1 (ApoA1), the major protein component of the serum high-density lipoprotein (HDL) fraction. Consistent with this observation, purified HDL restored solubilization of the N-processed, bioactive Shh variant from washed cells. We also found that the most C-terminal cholesteroylated Shh peptide is sufficient for Disp-mediated protein transfer to HDL. In contrast, palmitoylated N-terminal Shh membrane anchors are not transferred by Disp, but undergo shedding to complete Shh transfer to HDL. These results unify previously disparate models of Disp-, sheddase-, and LPP-mediated Shh solubilization into a comprehensive system that is fully consistent with published reports on in vivo Disp functions (*Burke et al., 1999*; *Ma et al., 2002*; *Callejo et al., 2011*; *Kawakami et al., 2002*; *Nakano et al., 2004*), in vivo Scube2 functions (*Hollway et al., 2006*; *Johnson et al., 2012*; *Kawakami et al., 2005*), the in vitro role of the Scube2 CUB domain (*Jakobs et al., 2017*; *Jakobs et al., 2014*), the Disp structure (*Chen et al., 2020*; *Li et al., 2021*), and required N-terminal – but not C-terminal – shedding

during in vivo Hh solubilization (*Kastl et al., 2018*; *Schürmann et al., 2018*; *Manikowski et al., 2020*). The results are also consistent with the established importance of Hh C-cholesterol for Hh association into 'large punctate' structures visible by light microscopy that may represent Hh-LPP complexes (*Gallet et al., 2003*; *Gallet and Therond, 2005*; *Gallet et al., 2006*), C-cholesterol-dependent Hh spreading (*Li et al., 2006*; *Tukachinsky et al., 2012*), and previously established in vivo roles of LPPs in Hh biofunction (*Eugster et al., 2007*; *Panáková et al., 2005*; *Palm et al., 2013*). We propose that the Disp-produced, HDL-bound, mono-lipidated Shh variant presented in this work helps to meet the requirements for regulating Hh activity in specific cell types and developing tissues.

## Results

### Synergistic function of Disp and Scube2 increases the shedding of dual-lipidated Shh from the plasma membrane

To analyze Shh solubilization from the plasma membrane, we produced dual-lipidated, tightly plasma membrane-associated morphogens in Bosc23 cells that endogenously express Disp (*Jakobs et al., 2014*) (referred to in this study as CRISPR non-targeting control [nt ctrl] cells) and in Bosc23 cells made deficient in Disp function by CRISPR/Cas9 (using previously characterized Disp$^{-/-}$ cells [*Ehring et al., 2022*]; experimental outlines and specificity controls are shown in *Figure 1—figure supplement 1*). Shh biosynthesis in both cell lines begins with the covalent attachment of a cholesterol moiety to the C-terminus of Shh (*Bumcrot et al., 1995*). This reaction is autocatalytic and tightly coupled to the generation of 19 kDa Shh signaling domains from a 45 kDa precursor. However, the subsequent N-palmitoylation of cholesteroylated Shh/Hh requires a separate enzymatic activity encoded by Hhat (*Chamoun et al., 2001*). Since Bosc23 cells do not express endogenous Hhat (*Jakobs et al., 2014*), we minimized the production of non-palmitoylated Shh by using bicistronic mRNA encoding the Shh precursor together with Hhat. In contrast to Shh expression in the absence of Hhat, Shh/Hhat expression ensures nearly quantitative Shh N-palmitoylation in transfected Bosc23 cells (*Jakobs et al., 2014*). SDS-PAGE/immunoblotting was then used to characterize dual-lipidated Shh release from the plasma membrane of both cell lines into serum-depleted media. This confirmed that Scube2 enhances Shh solubilization from nt ctrl cells (*Figure 1A*, arrowhead; *Figure 1—figure supplement 2A* includes all transfection and loading controls) and that Shh solubilization from Disp$^{-/-}$ cells is always strongly impaired (*Figure 1A'*; *Ehring et al., 2022*; *Tukachinsky et al., 2012*; *Creanga et al., 2012*). We also confirmed that the electrophoretic mobility of most Shh released from nt ctrl cells (*Figure 1A*, arrowhead) was increased over that of the corresponding dual-lipidated cellular precursor (*Figure 1A*, asterisk; *Jakobs et al., 2014*; *Jakobs et al., 2016*; *Ehring et al., 2022*). RP-HPLC of the solubilized material shown in *Figure 1A* (arrowhead) demonstrated that the observed increase in electrophoretic mobility was caused by the proteolytic removal of both lipidated terminal peptides from the cellular protein (*Figure 1A''*, this post-translational modification of Shh is referred to as shedding throughout this article; see *Figure 1—figure supplement 2B–G* for cellular dual-lipidated or artificially produced monolipidated and unlipidated Shh standard proteins and see *Ehring et al., 2022*, which rules out alternative modes of Shh deacylation). Transgenic Disp expression in Disp$^{-/-}$ cells restored Shh shedding and solubilization, confirming the specificity of the assay (*Figure 1—figure supplement 2H*; *Ehring et al., 2022*). Strikingly, the same phenotypic reversal was achieved by the coexpression of the cholesterol pump Patched (Ptch1) (*Zhang et al., 2018*), which depletes the plasma membrane of free sterols (*Figure 1—figure supplement 2H*; *Ehring et al., 2022*). This suggests an indirect 'second messenger' role of plasma membrane cholesterol not only in the regulation of Smoothened downstream of Ptch1 in Shh receiving cells (*Zhang et al., 2018*; *Kinnebrew et al., 2019*; *Kinnebrew et al., 2022*), but also in the regulation of Shh release by Disp in producing cells (*Ehring et al., 2022*). Taken together, Scube2 and Disp synergistically and specifically increased Shh release via shedding from the cell surface (as indicated by the appearance of 'lower'-sized bands compared to those of cellular proteins, *Figure 1A*, and the decreased hydrophobicity of solubilized Shh, *Figure 1A''*).

We also observed that solubilization of engineered monolipidated Shh variants (${}^{C25S}$Shh, lacking N-palmitate, *Figure 1B and B'*, and ShhN, lacking C-cholesterol, *Figure 1C and C'*, *Figure 1—figure supplement 2I and J*) remained linked with the shedding of the respective lipidated membrane termini, as shown by electrophoretic mobility shifts of soluble proteins (*Figure 1B and C*, arrowheads) and their delipidation during solubilization (as determined by RP-HPLC, *Figure 1B'' and C''*).

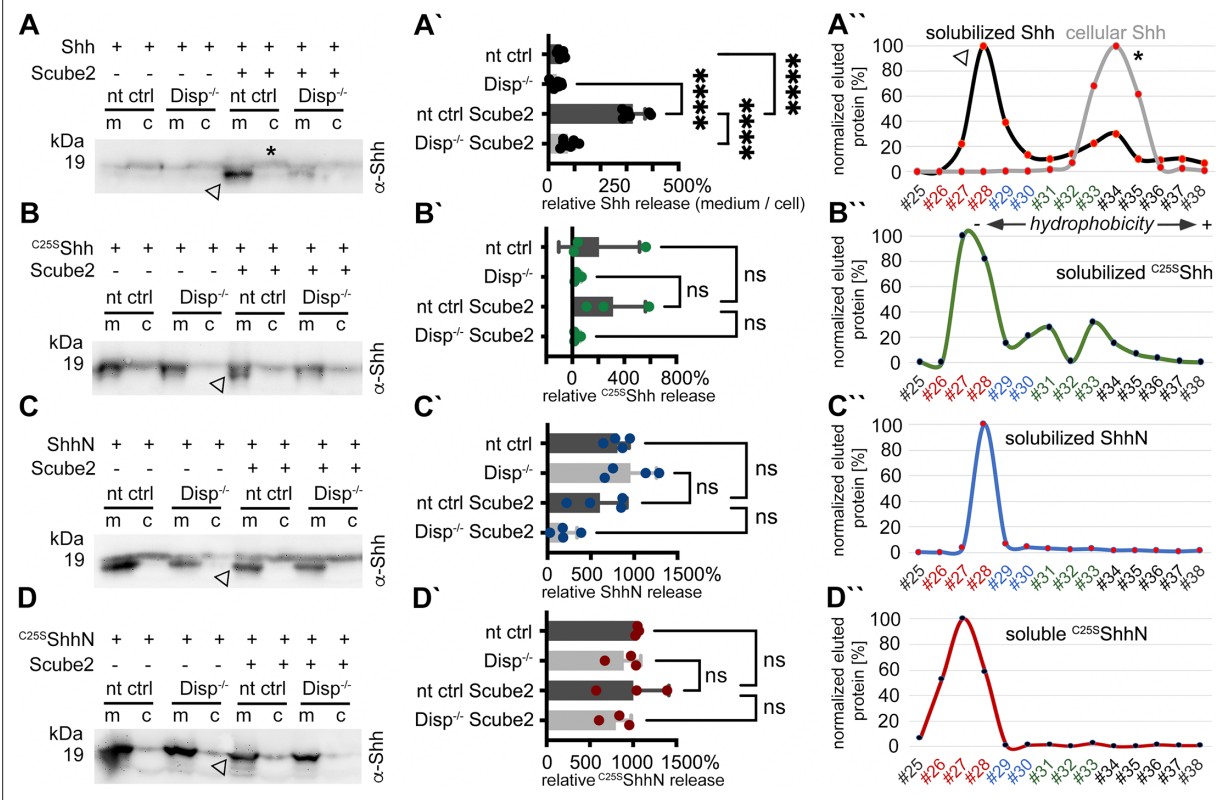

**Figure 1.** Disp and Scube2 enhance the shedding of cell surface-associated dual-lipidated Shh into delipidated soluble forms. Media containing 10% serum was changed to serum-free Dulbecco's Modified Eagle's Medium (DMEM) 36 hr post-transfection and proteins were solubilized for 6 hr. Cells were not washed between media changes to leave residual serum traces in the assay. We refer to this experimental condition as 'serum-depleted' throughout this article. (**A**) Cap-dependent Shh translation and cap-independent Hhat translation from bicistronic mRNA ensured the generation of dual-lipidated, plasma membrane-associated proteins (asterisk in the cellular fraction) in all transfected cells. Disp and Scube2 synergistically and specifically enhance the conversion of dual-lipidated Shh precursors into truncated soluble variants during release (arrowhead). m: media; c: cell lysate. (**A'**) Quantification of relative Shh release from non-targeting control (nt ctrl) and Disp$^{-/-}$ cells in the presence or absence of Scube2. Amounts of solubilized Shh with higher electrophoretic mobility (the lower bands) were quantified and expressed as % relative to the respective cellular Shh, which was always set to 100%. One-way ANOVA, Dunnett's multiple-comparisons test. ****p<0.0001. See **Supplementary file 1** for detailed statistical information. (**A''**) Reverse-phase high-performance liquid chromatography (RP-HPLC) analyses revealed that Shh solubilized by Disp and Scube2 (the same fraction indicated by the arrowhead in **A**, black line) was less hydrophobic than its cell surface-associated precursor (gray line; the asterisk indicates analysis of the same cellular fraction as shown in **A**). RP-HPLC calibration and color coding of Shh fractions are shown in **Figure 1—figure supplement 2B–G**. (**B–D**) Solubilization of non-palmitoylated $^{C25S}$Shh (in this artificial variant, the N-terminal palmitate acceptor cysteine is replaced by a non-accepting serine; functionally equivalent constructs with the cysteine exchanged for a non-accepting alanine [$^{C25A}$Shh] were also used in our study), non-cholesteroylated but palmitoylated ShhN and lipid-free control $^{C25S}$ShhN under the same serum-depleted conditions. Arrowheads indicate Shh variants that were solubilized in Disp- and Scube2 presence. (**B'–D'**) Processed protein quantifications from (**B–D**), again from nt ctrl and Disp$^{-/-}$ cells in the presence or absence of Scube2. One-way ANOVA, Dunnett's multiple-comparisons test. ns: p>0.05. See **Supplementary file 1** for detailed statistical information. (**B''–D''**) RP-HPLC shows similar elution of $^{C25S}$Shh, ShhN, and non-lipidated $^{C25S}$ShhN. This indicates that terminal lipids were removed during the release of $^{C25S}$Shh and ShhN (as well as Shh, **A''**).

The online version of this article includes the following source data and figure supplement(s) for figure 1:

**Source data 1.** Raw data and statistical analyses of **Figure 1**.

**Figure supplement 1.** Experimental design and controls.

**Figure supplement 1—source data 1.** Raw data of **Figure 1—figure supplement 1**.

**Figure supplement 2.** Loading controls, activity controls, and standards on the same stripped blots.

**Figure supplement 2—source data 1.** Uncropped western blots of **Figure 1—figure supplement 2**.

However, their release was much less controlled than that of dual-lipidated Shh (**Figure 1B' and C'**), consistent with the observations of others (**Tukachinsky et al., 2012**; **Creanga et al., 2012**). An unprocessed protein fraction, represented by the 'top' bands on all immunoblots, was also released in a manner that was independent of Disp and Scube2 (**Figure 1—figure supplement 2K–M**), as was

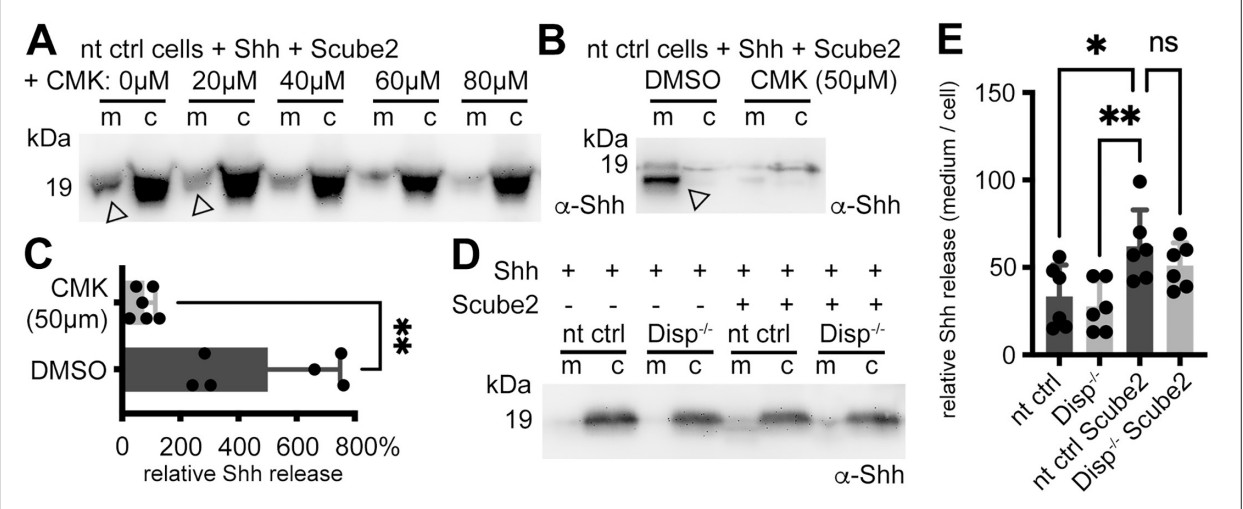

**Figure 2.** Shh shedding depends on Disp activation by furin and the presence of serum traces. (**A**) Non-targeting control (nt ctrl) cells were transfected with Shh and Scube2, and Shh solubilization was monitored in the presence or absence of the furin inhibitor chloromethylketone (CMK). CMK impaired proteolytic processing and release of truncated (arrowheads) soluble Shh in a concentration-dependent manner. (**B**) Truncated (arrowhead) Shh release in the presence or absence of 50 μM CMK furin inhibitor. (**C**) Quantification of CMK-inhibited Shh shedding. Ratios of solubilized versus cellular Shh (set to 100%) in the presence of 50 μM CMK inhibitor were determined and expressed relative to Shh solubilization in the absence of inhibitor (DMSO). Unpaired *t*-test, two-tailed. **p=0.0021, n = 6. See *Figure 2—figure supplement 1A* for loading controls and *Supplementary file 1* for additional statistical information. (**D**) Immunoblotted cellular (c) and medium (m) fractions of Shh expressing nt ctrl and Disp⁻/⁻ cells in the complete absence of serum (referred to as 'serum-free' conditions throughout this article). Note that Shh solubilization is greatly reduced under serum-free conditions. (**E**) Processed Shh quantifications after secretion from nt ctrl and Disp⁻/⁻ cells in the presence or absence of Scube2 into serum-free medium. One-way ANOVA, Dunnett's multiple-comparisons test. **p=0.0059, *p=0.02, ns: p=0.54. See *Supplementary file 1* for detailed statistical information.

The online version of this article includes the following source data and figure supplement(s) for figure 2:

**Source data 1.** Raw data and statistical analyses of *Figure 2*.

**Figure supplement 1.** Loading controls.

**Figure supplement 1—source data 1.** Uncropped western blots of *Figure 2—figure supplement 1*.

the non-lipidated ᶜ²⁵ˢShhN control (*Figure 1D–D"*, *Figure 1—figure supplement 2N*). These results suggest that cell surface shedding represents a 'ground state' from which only dual-lipidated Shh is protected, until Disp and Scube2 render it susceptible to shedding. The results also suggest that dual N- and C-terminal Hh lipidation during biosynthesis serves to prevent unregulated protein release from producing cells. This highlights the importance of coupled Shh/Hhat expression in vitro to reliably characterize the mechanism of Disp- and Scube2-regulated Shh solubilization.

## Shh shedding is dependent on cleavage-activated Disp and the presence of serum

It was recently shown that the prohormone convertase furin cleaves Disp at a conserved processing site to activate it and to release Shh from the cell surface (*Stewart et al., 2018*). Based on this mode of activation, we hypothesized that furin inhibition might specifically interfere with Disp-regulated Shh shedding. To test this hypothesis, we added 0–80 μM peptidyl chloromethylketone (CMK, a competitive inhibitor of furin) to our solubilization assays. Indeed, CMK reduced Shh shedding from the cell surface in a concentration-dependent manner (*Figure 2A–C*, *Figure 2—figure supplement 1A*). During these assays, we also found that repeated careful prior washing of cells to quantitatively remove all traces of serum severely impaired Disp- and Scube2-mediated Shh solubilization into serum-free media (*Figure 2D and E*, *Figure 2—figure supplement 1B*). From this latter observation, we derive two important conclusions. The first is that the minimal requirements of Na⁺-driven, Disp-mediated Shh extraction and hand-over to Scube2 (*Wang et al., 2021*) are not sufficient to release Shh. The second is that Shh self-assembly by the law of mass action (*Koleva et al., 2015*) is also not supported because this process should solubilize Shh regardless of the presence or absence of serum. In contrast, we previously found that one function of Disp is to extract free plasma membrane

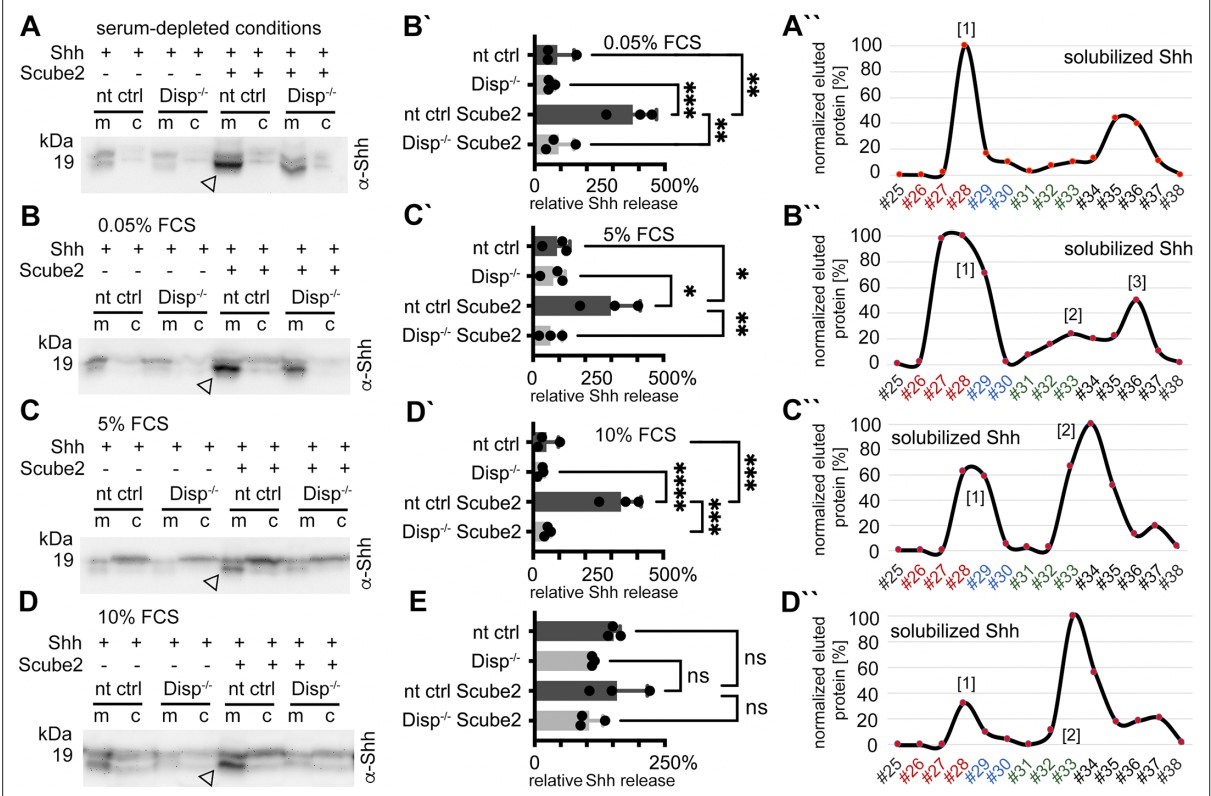

**Figure 3.** Dually lipidated cell surface Shh converts to delipidated soluble forms under low-serum and high-serum conditions. Media were changed to serum-free Dulbecco's Modified Eagle's Medium (DMEM) (cells were not washed) or DMEM containing the indicated amounts of serum 36 hr post-transfection, and proteins were solubilized for 6 hr (serum-depleted) or for 24 hr (with 0.05, 5, and 10% serum). (**A**) Under serum-depleted conditions, Disp and Scube2 increase the conversion of dual-lipidated Shh into truncated soluble forms (arrowhead). m: media; c: cell lysate. (**A″**) Reverse-phase high-performance liquid chromatography (RP-HPLC) confirmed the loss of both terminal lipidated Shh peptides during Disp- and Scube2-regulated shedding, as shown previously (**Figure 1A″**). (**B–D**) The appearance of truncated Shh in serum-containing media remained dependent on Disp and, to a lesser extent, Scube2 (arrowheads). (**B′–D′**) Quantifications of (**B–D**). One-way ANOVA, Dunnett's multiple-comparisons test. ****$p<0.0001$, ***$p<0.001$, **$p<0.01$, *$p<0.05$. See **Supplementary file 1** for detailed statistical information. (**B″–D″**) RP-HPLC revealed that increased serum levels shift dual Shh shedding [1] toward N-terminally restricted shedding and release of a cholesteroylated Shh form [2]. Low levels of dually lipidated Shh are also detected [3]. (**E**) Quantification of unprocessed Shh release in the presence of 10% fetal calf serum (FCS) (top band in **D**). One-way ANOVA, Dunnett's multiple-comparisons test. ns: $p>0.05$. See **Figure 3—figure supplements 1 and 2** for additional information.

The online version of this article includes the following source data and figure supplement(s) for figure 3:

**Source data 1.** Raw data and statistical analyses for **Figure 3**.

**Figure supplement 1.** Loading controls.

**Figure supplement 1—source data 1.** Uncropped western blots for **Figure 3—figure supplement 1**.

**Figure supplement 2.** Source data and reverse-phase high-performance liquid chromatography (RP-HPLC) profiles.

**Figure supplement 2—source data 1.** Uncropped western blots for **Figure 3—figure supplement 2**.

cholesterol and transfer it to a soluble sink for removal from the cell (**Ehring et al., 2022**; **Ehring and Grobe, 2021**). In vertebrates, HDLs represent a soluble sink for 'free' peripheral cholesterol (**Luo et al., 2020**), and HDLs and the pharmacological cholesterol chelator methyl-β-cyclodextrin have previously been shown to increase Shh shedding (**Ehring et al., 2022**; **Jakobs et al., 2014**; **Ohlig et al., 2012**). Taken together, these results suggest that the permissive factor missing from our shedding assay is most likely a soluble cholesterol acceptor, such as HDL or a related serum LPP, as previously reported (**Palm et al., 2013**).

## Elevated serum concentrations shift shedding of both Shh termini to selective N-terminal shedding

To further characterize serum-dependent shedding, we expressed dual-lipidated Shh in nt ctrl cells and Disp$^{-/-}$ cells and solubilized the proteins into serum-depleted Dulbecco's Modified Eagle's Medium (DMEM) or into DMEM supplemented with 0.05, 5, and 10% fetal calf serum (FCS) (*Figure 3A–D*, quantification of release in *Figure 3B´-D´*; *Figure 3—figure supplement 1A–D* shows transfection and loading controls). Consistent with previous observations, we again found that Scube2 enhanced Shh shedding from Disp-expressing cells into serum-depleted media (*Figure 3A*, arrowhead), but did not significantly enhance Shh shedding from Disp$^{-/-}$ cells. RP-HPLC of the solubilized material confirmed that the observed increase in electrophoretic mobility was caused by the proteolytic removal of both terminal lipidated peptides (*Figure 3A"*). Increased serum concentrations during Shh solubilization (0.05% FCS [*Figure 3B*, quantified in *Figure 3B'*], 5% FCS [*Figure 3C*, quantified in *Figure 3C'*], and 10% FCS [*Figure 3D*, quantified in *Figure 3D'*]) did not appear to much affect Disp- and Scube2-specific Shh shedding (arrowheads, *Figure 3E* in contrast shows unregulated release of unprocessed Shh [the 'top' band] in the presence of 10% FCS). To our surprise, however, RP-HPLC of the solubilized materials shown in *Figure 3B–D* revealed a gradual shift from dual Shh shedding (labeled [1] in *Figure 3A"–D"*) toward the solubilization of variant proteins with their C-terminal cholesteroylated peptide still intact (labeled [2] in *Figure 3B"–D"*, *Figure 3—figure supplement 2A–E*). We also observed this hydrophobicity shift (together with an electrophoretic mobility shift, see *Figure 3—figure supplement 1D*) when analyzing endogenous Shh from the pancreatic cancer cell line Panc1 (*Figure 3—figure supplement 2F–H*). We note that small amounts of dual-lipidated Shh were again present (indicated by asterisks in *Figure 3—figure supplement 2B–E*). However, the relative amounts of this protein fraction increased both in the absence of Scube2 (*Figure 3—figure supplement 2I*; here, cells had to be incubated for 24 hr to compensate for the low Shh solubilization) and in the absence of Disp (*Figure 3—figure supplement 2J*). In contrast, Scube2 expression in Disp$^{-/-}$ cells increased the relative amount of delipidated soluble Shh (*Figure 3—figure supplement 2K*). From this, we conclude that the Disp- and Scube2-independent solubilization of dual-lipidated Shh is physiologically irrelevant, consistent with the findings shown in *Figure 1—figure supplement 2K–M*.

So far, our data have shown that Scube2 increases Shh shedding only from Disp-expressing cells and that serum enhances this process, probably by providing a sink for membrane cholesterol transferred by Disp (*Ehring et al., 2022*; *Figure 3—figure supplement 2L*). We also showed that high serum levels promote a second Disp solubilization mode in which Shh shedding is restricted to the palmitoylated N-peptide, leaving the C-terminus intact. This latter finding raised the interesting possibility that serum factors may accept or protect the cholesteroylated Shh C-terminus. Indeed, it is known that a filtrate of blood serum through the capillary walls, called interstitial fluid, represents the microenvironment of tissues and cells in vivo, as well as of tissues and cells that express and solubilize Shh during development (*Palm et al., 2013*). It is also known that the interstitial fluid is rich in LPPs of small mass from the serum (*Lundberg et al., 2013*). This suggests that Shh expression in the presence of serum resembles the conditions in Shh-expressing tissues in vivo and may thus be physiologically relevant.

## Shh-induced in vitro differentiation of C3H10T1/2 cells and activation of NIH3T3 cells do not require N-palmitate

Is this novel N-terminally processed Shh variant functional? It is well established that dual lipidation during biosynthesis is absolutely necessary for unimpaired Hh/Shh biofunction in vivo (*Gallet et al., 2006*; *Porter et al., 1996a*; *Lewis et al., 2001*; *Huang et al., 2007*; *Lee et al., 2001*). These studies have shown that dual-lipidated Hh/Shh expression generates soluble variants that are 10–30 times more bioactive than engineered proteins that do not undergo Hhat-catalyzed N-palmitoylation but undergo unperturbed C-terminal cholesteroylation and secretion to the cell surface (*Chamoun et al., 2001*; *Lee et al., 2001*). According to these published observations, the N-terminally processed Shh variant described here should not be very active because it lacks the palmitate. To test this hypothesis, we used the Ptch1-expressing multipotent fibroblastic cell line C3H10T1/2 as a reporter (*Nakamura et al., 1997*). We first verified the multipotency of our C3H10T1/2 cells to differentiate into osteoblasts (*Nakamura et al., 1997*), chondrocytes (*Wa et al., 2017*), or adipocytes (*Tang et al., 2004*; *Figure 4—figure supplement 1A*). To this end, C3H10T1/2 cells were cultured in the presence

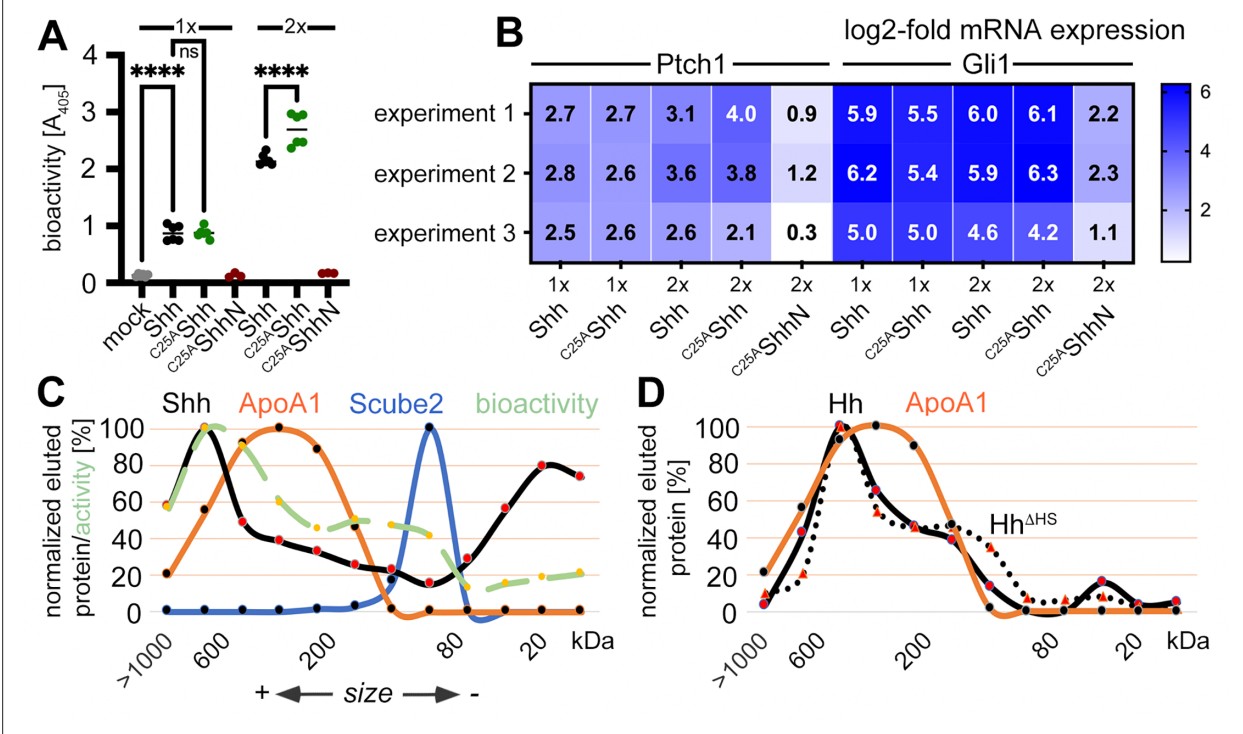

**Figure 4.** Activities and size-exclusion chromatography (SEC) of dual-lipidated Shh/Hh and depalmitoylated Shh variants solubilized into serum-containing media. (**A**) Shh, $^{C25A}$Shh (this artificial variant has the cysteine palmitate acceptor changed to a non-accepting alanine), and non-lipidated $^{C25A}$ShhN were expressed in media containing 10% fetal calf serum (FCS); their protein levels were determined by immunoblotting and normalized; and the conditioned media were added to C3H10T1/2 reporter cells to induce their Hh-dependent differentiation into alkaline phosphatase (Alp)-producing osteoblasts. Mock-treated C3H10T1/2 cells served as non-differentiating controls. At lower concentrations (1×), Shh and $^{C25A}$Shh induced C3H10T1/2 differentiation in a similar manner, as determined by Alp activity measured at 405 nm. At higher concentrations (2×), the bioactivity of $^{C25A}$Shh was increased over that of Shh. $^{C25A}$ShhN was inactive. One-way ANOVA, Sidak's multiple-comparisons test. ****p<0.0001, ns = 0.99, n = 3–9. See **Supplementary file 1** for additional statistical information. (**B**) Similar transcription of the Hh target genes Ptch1 and Gli1 by Shh and $^{C25A}$Shh in three independent experiments. C3H10T1/2 reporter cells were stimulated with similar amounts of Shh, $^{C25A}$Shh, and $^{C25A}$ShhN (**Figure 4—figure supplement 1E**) at high (2×) and low (1×) concentrations. (**C**) SEC shows significant amounts of Shh of increased molecular weight in media containing 10% serum (black line). The increased molecular weight Shh eluted together with ApoA1 (orange line). In contrast, Scube2 was largely monomeric in solution (blue line). The level of Shh-induced Alp activity in C3H10T1/2 cells was measured as absorbance at 405 nm, showing the strongest C3H10T1/2 differentiation by eluted fractions containing large Shh aggregates. (**D**) SEC of *Drosophila* Hh (black line) and of a variant lacking its HS binding site (Hh$^{\Delta HS}$, black dotted line). Both proteins were expressed from S2 insect cells under actin-Gal4/UAS-control and solubilized into media containing 10% FCS.

The online version of this article includes the following source data and figure supplement(s) for figure 4:

**Source data 1.** Raw data of **Figure 4**.

**Figure supplement 1.** Confirmed multipotency of C3H10T1/2 cells and similar activities of palmitoylated and non-palmitoylated Shh released into serum containing media.

**Figure supplement 1—source data 1.** Raw data for bioassays shown in **Figure 4—figure supplement 1**.

of adipogenic, chondrogenic, and osteogenic supplements for different periods of time, and their responsiveness was confirmed based on phenotype and the expression of cell surface markers. We then incubated C3H10T1/2 cells with Shh expressed in the presence of 10% serum and the physiological release regulators Scube2 (**Johnson et al., 2012**; **Kawakami et al., 2005**) and Disp (**Burke et al., 1999**; **Ma et al., 2002**). Shh shares 91% sequence identity and both lipids with Indian hedgehog, an established osteogenic factor (**Pathi et al., 2001**), and both Indian hedgehog and Shh stimulate C3H10T1/2 osteogenic differentiation. We confirmed that Shh induced alkaline phosphatase (Alp) expression and C3H10T1/2 differentiation into osteoblasts in a concentration-dependent manner (**Nakamura et al., 1997**; **Figure 4A**). Notably, we observed that $^{C25A}$Shh expressed under the same conditions was as active or even slightly more active than Shh. In contrast, the biofunction of the $^{C25S}$ShhN negative control was always very low. Quantitative reverse transcription-polymerase chain reaction (qPCR) analysis of Ptch1 and Gli1 expression in Shh-treated C3H10T1/2 cells (**Figure 4B**)

or Shh-treated NIH3T3 cells (*Figure 4—figure supplement 1B*) confirmed this finding: after protein normalization, qPCR confirmed similar increases in Ptch1 and Gli1 mRNA expression under direct Shh or [C25A]Shh control. This is consistent with Ptch1 being known to be upregulated by Shh (*Marigo and Tabin, 1996*), and Gli1 is a zinc finger transcription factor that acts downstream of Ptch1 and is also transcribed in an Hh-dependent manner (*Lee et al., 1997*). Key regulators of adipogenesis, osteogenesis, chondrogenesis, and proliferation in Shh and [C25S]Shh-induced C3H10T1/2 cells were also tested and remained similar (*Figure 4—figure supplement 1C and D*, *Supplementary files 2 and 3*). Finally, we incubated C3H10T1/2 cells with Shh R&D 8908-SH, a commercially available dual-lipidated Shh variant obtained by detergent extraction from transfected cells (*Figure 4—figure supplement 1E and F*). R&D 8908-SH induced Alp expression in differentiating C3H10T1/2 cells in a concentration-dependent manner, as expected (*Nakamura et al., 1997*). Importantly, the activities of solubilized Shh and [C25A]Shh were increased over similar amounts of R&D 8908-SH (*Figure 4—figure supplement 1F*), and qPCR of Ptch1 and Gli1 expression confirmed similar activities of R&D 8908-SH, Shh, and [C25A]Shh (*Figure 4—figure supplement 1G*). These results demonstrate that N-palmitate is not essential for the strength of Shh signaling at the level of Ptch1 when [C25A]Shh is released in the presence of serum.

## Soluble Shh/Hh and ApoA1-containing LPPs have a similar size

A decade ago, it was already known that flies and mammals release sterol-modified Hh/Shh in LPP-associated bioactive form as well as in a desteroylated, unassociated form (*Palm et al., 2013*). These results suggest that serum LPPs in our assays may not only have promoted Shh shedding (*Figures 1 and 3*; *Ehring et al., 2022*), but may also have promoted assembly of monolipidated Shh into soluble LPP-associated complexes. We used SEC to test this possibility. SEC detects soluble Shh monomers (20 kDa) and covers the entire molecular weight (MW) range up to a cutoff of about $10^6$ Da. The MW range of serum HDL is from $1.75 \times 10^5$ Da to $3.6 \times 10^5$ Da, corresponding to particle sizes of 5–11 nm. This small size range makes HDLs abundant components of interstitial fluid and good candidates for Shh solubilization from cells and tissues. Larger complexes in mammalian serum include low-density lipoproteins (LDLs, MW $2.75 \times 10^6$ Da) and very low-density lipoproteins (VLDLs, MW $10–80 \times 10^6$ Da), both of which would elute in the void column volume. As shown in *Figure 4C*, Shh in DMEM + 10% FCS elutes in fractions covering the entire MW range (black line). A prominent Shh peak is detected at around 20 kDa – most likely representing the dually cleaved, fully delipidated Shh fraction (as shown in *Figure 3A"–D"*, labeled [1]) – and a second peak is detected between 300 kDa and 600 kDa. Reprobing the same (stripped) blot with antibodies directed against ApoA1 (the major protein component of HDL) revealed an elution profile that overlapped with that of Shh (*Figure 4C*, orange line). The leftward shift of Shh elution relative to HDL elution may be explained by the increased size of an LPP subfraction after its Disp-mediated loading with Shh. In contrast, Scube2 (*Figure 4C*, blue line) co-eluted only with smaller Shh multimers (*Koleva et al., 2015*). To determine which size fraction contained the biologically active Shh, we analyzed the differentiation of C3H10T1/2 osteoblast progenitor cells by using aliquots of eluted fractions from the same SEC run. We found that Shh induced C3H10T1/2 osteoblast differentiation with a size-dependent activity distribution: large Shh assemblies were highly bioactive, smaller assemblies were also bioactive, but monomeric Shh was only moderately active (*Figure 4C*, green dashed line). We also found that *Drosophila* Hh expressed in *Drosophila* S2 cells and solubilized into serum-containing medium showed a similar size distribution (*Figure 4D*, black line) and that Hh assemblies also co-eluted with serum ApoA1 (orange line). This suggests that the mechanism of serum-enhanced Hh assembly during solubilization is conserved. The alternative possibility of Hh assembly as a consequence of heparan sulfate (HS)-proteoglycan interactions (*Whalen et al., 2013*) was rejected because site-directed mutagenesis of the HS-binding Cardin–Weintraub motif did not alter the size distribution of Hh[ΔHS] (*Figure 4D*, black dotted line). These results confirm a previous study showing that *Drosophila* Hh copurifies with LPP and co-localizes with LPP in the developing wing epithelium in vivo (*Panáková et al., 2005*). The same study showed that reduced LPP levels in larvae lead to Hh accumulation at the site of production due to impaired Hh release, similar to the in vitro results shown in *Figure 2D and E*.

## Disp function requires HDL

To test whether the soluble LPP that enhances Disp-mediated solubilization of N-processed, cholesteroylated Shh is HDL, we analyzed this possibility directly. To this end, we expressed Shh in nt ctrl

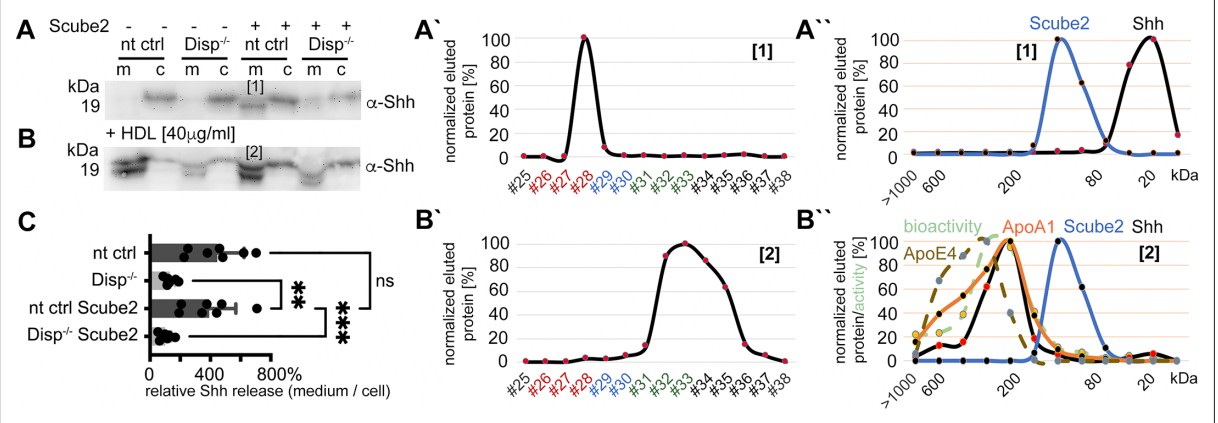

**Figure 5.** High-density lipoprotein (HDL) enhances N-processed Shh solubilization by Disp. Media were changed to serum-free Dulbecco's Modified Eagle's Medium (DMEM) (and cells washed three times) or serum-free DMEM supplemented with 40 μg/mL HDL 36 hr post-transfection before protein solubilization for 6 hr. (**A**) Immunoblotted cellular (c) and medium (m) fractions of Shh expressing nt ctrl and Disp[-/-] cells in the complete absence of serum. Note that only little Shh is released under this condition [1]. (**A'**) Reverse-phase high-performance liquid chromatography (RP-HPLC) of the material labeled [1] in (**A**) showed complete delipidation during release. (**A''**) Size-exclusion chromatography (SEC) of the same delipidated material shows that it is readily soluble and not associated with Scube2. (**B**) Immunoblotted cellular (c) and medium (m) fractions of Shh expressing nt ctrl and Disp[-/-] cells in the presence of 40 μg/mL HDL. Shh shedding and solubilization are strongly increased by Disp [2] but not by Scube2. (**B'**) RP-HPLC of the material labeled [2] in (**B**) showed that HDL shifts Shh shedding from dual processing (**A'**, [1]) to release of cholesteroylated Shh. (**B''**) SEC of the same material [2] (black line) shows an increase in molecular weight corresponding to the molecular weight range of HDL, as indicated by the marker apolipoproteins ApoA1 (orange line) and mobile ApoE4. The former provides structural stability to the particle and stimulates cholesterol efflux to HDL; the latter facilitates cholesterol storage and core expansion and is therefore a marker of larger mature HDL particles (brown dashed line). Again, the soluble Shh elution profile does not overlap with that of Scube2 (blue line). (**C**) Quantification of HDL-induced Shh solubilization from nt ctrl cells and Disp[-/-] cells. One-way ANOVA, Dunnett's multiple-comparisons test. ***p=0.0008, **p=0.0023, ns: p=0.77, n = 7. Additional statistical information is provided in **Supplementary file 1**.

The online version of this article includes the following source data and figure supplement(s) for figure 5:

**Source data 1.** Raw data of **Figure 5**.

**Figure supplement 1.** Loading controls.

**Figure supplement 1—source data 1.** Raw data of **Figure 5—figure supplement 1**.

cells and Disp[-/-] cells in the presence or absence of Scube2, washed the cells 36 hr after transfection to inhibit Shh release, and added serum-free DMEM or DMEM supplemented with 40 μg/mL purified human HDL. We first confirmed that the absence of serum factors greatly reduced Shh solubilization (**Figure 5A**, the released Shh is labeled [1], **Figure 5—figure supplement 1A**). RP-HPLC of the small amounts of released Shh confirmed the removal of both lipidated Shh peptide anchors during solubilization (**Figure 5A'**, **Figure 5—figure supplement 1A'**), and SEC confirmed that the solubilized proteins were monomeric and not Scube2-associated (**Figure 5A''**). In contrast, Shh solubilization from nt ctrl cells into DMEM supplemented with 40 μg/mL HDL was greatly increased (**Figure 5B and C**, **Figure 5—figure supplement 1B**). As previously observed for Shh expression in the presence of serum, RP-HPLC of HDL-solubilized Shh (**Figure 5B**, labeled [2]) confirmed that its C-terminal cholesteroylated peptides were still intact (**Figure 5B'**, **Figure 5—figure supplement 1B'**) and that the proteins were highly bioactive (**Figure 5B''**, green dashed line). Solubilized dual-lipidated Shh – represented by the 'upper' bands – was also found, but its solubilization was again independent of Scube2 and Disp function (**Figure 5—figure supplement 1B'' and B'''**), indicating a non-specific release. From the observed size shift and overlap with ApoA1 and ApoE4 elution profiles in SEC (**Figure 5B''**, note that the observed Shh MW range almost matches that of ApoA1), we suggest that monolipidated truncated Shh associates with HDL in a Disp-regulated manner. This possibility is supported by the previous observation that Hh levels were greatly decreased in the hemolymph of Disp-deficient fly larvae (**Palm et al., 2013**). We also tested whether HDL is the only possible Shh acceptor, or whether LDL can also carry monolipidated morphogens. As shown in **Figure 5—figure supplement 1C and D**, Shh was transferred to HDL-supplemented serum-free DMEM, but not to LDL-supplemented

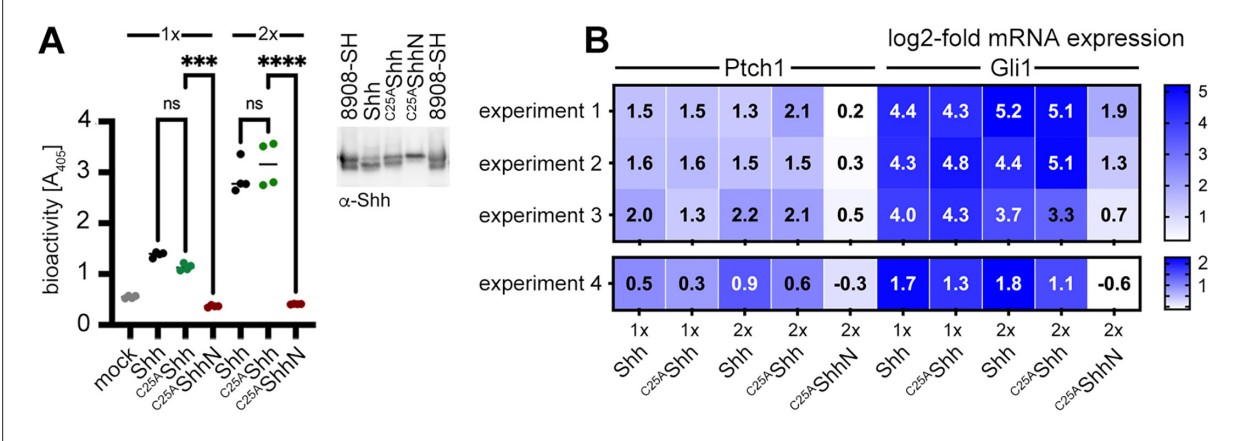

**Figure 6.** Activities of high-density lipoprotein (HDL)-associated Shh and non-palmitoylated variants. (**A**) Shh, non-palmitoylated [C25A]Shh, and non-lipidated [C25A]ShhN were released into media containing 80 μg/mL HDL, their protein levels were determined by immunoblotting and normalized (inset), and conditioned media were added to C3H10T1/2 reporter cells to induce their differentiation into Alp-producing osteoblasts. Mock-treated C3H10T1/2 cells served as non-differentiating controls. At low (1×) and high (2×) concentrations, Shh and [C25A]Shh induced C3H10T1/2 differentiation in a similar manner, as determined by Alp activity measured at 405 nm. Again, [C25A]ShhN was completely inactive, in contrast to the bioactive HDL-associated non-palmitoylated [C25S]Shh. One-way ANOVA, Sidak's multiple-comparisons test. ****p<0.0001, ***p<0.001, ns > 0.1, n = 4. Additional statistical information is provided in *Supplementary file 1*. (**B**) Similar transcription of Hh target genes Ptch1 and Gli1 by HDL-associated Shh and [C25A]Shh in C3H10T1/2 cells three independent experiments. Experiment 4 confirms similar Shh and [C25A]Shh activities in NIH3T3 cells. Reporter cells were stimulated with similar amounts of Shh, [C25A]Shh, and [C25A]ShhN at high (2×) and low (1×) concentrations as determined by immunoblotting (**A**, inset).

The online version of this article includes the following source data for figure 6:

**Source data 1.** Statistical analyses of *Figure 6*.

serum-free DMEM. Shh released by HDL is N-terminally processed (*Figure 5—figure supplement 1G*) and physically interacts with the LPP (*Figure 5—figure supplement 1E and F*).

Is N-terminally processed, HDL-associated Shh functional? To answer this question, we again used NIH3T3 cells and the multipotent fibroblastic cell line C3H10T1/2 as a reporter. Shh was co-expressed with Scube2 in the presence of 80 μg/mL purified human HDL and the conditioned media was added to C3H10T1/2 cells. As expected, Shh/HDL complexes induced Alp expression and C3H10T1/2 differentiation into osteoblasts in a concentration-dependent manner (*Nakamura et al., 1997*) and [C25A]Shh expressed under the same conditions was equally active (*Figure 6A*). qPCR of Ptch1 and Gli1 expression in Shh/HDL-stimulated or [C25A]Shh/HDL-stimulated C3H10T1/2 cells and NIH3T3 cells confirmed this finding: both protein/HDL complexes induced similar increases in Ptch1 and Gli1 mRNA expression (*Figure 6B*). This suggests that the HDL-associated, N-terminally truncated Shh variants described in this study are highly bioactive.

## The C-terminal cholesterylated Shh peptide is necessary and sufficient for the Disp-mediated export of the protein and the association with HDL

We next investigated whether the C-cholesteroylated peptide is sufficient for Disp-mediated Shh transfer to HDL. This possibility was suggested by the loss of the N-palmitoylated peptide during solubilization from Disp-expressing cells in the presence of serum (*Figures 1A and 3A*) and by Disp-independent ShhN solubilization into serum-depleted medium (*Figure 1C*; *Tian et al., 2005*). We expressed palmitoylated/non-cholesterylated ShhN and non-palmitoylated/cholesterylated [C25S]Shh and confirmed that palmitoylated ShhN solubilization from the plasma membrane was independent of Disp and HDL (*Figure 7A*, [1] denotes material solubilized into serum-free DMEM, [2] denotes material solubilized by HDL; see also *Figure 7—figure supplement 1A and B*). SEC detected only monomeric ShhN, regardless of the presence or absence of HDL (*Figure 7A′ and A″*). These results suggest that the N-terminal palmitate of Shh is not a substrate for Disp-mediated transfer to HDL. In contrast, only small amounts of monomeric [C25S]Shh were solubilized into serum-free media in HDL absence (*Figure 7B and B′*, labeled [3]), and HDL strongly increased [C25S]Shh release from Disp-expressing cells

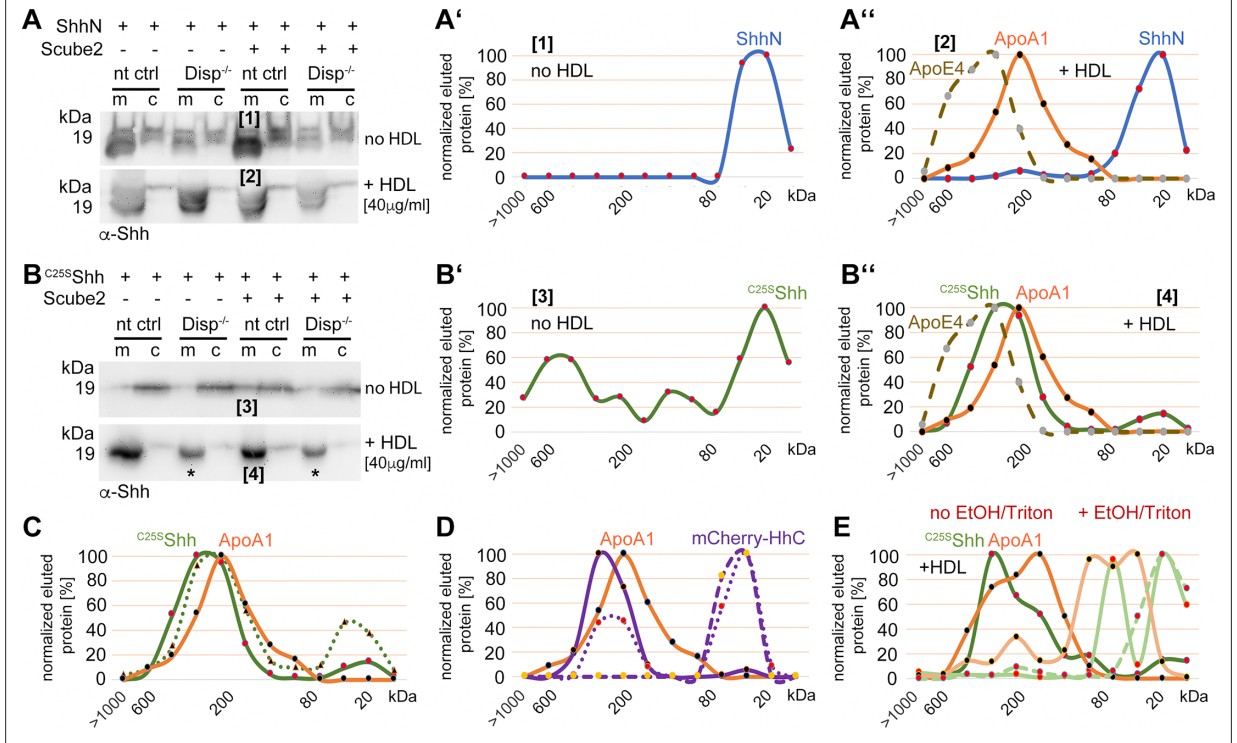

**Figure 7.** Cholesterylated C-terminal peptides are necessary and sufficient for Disp-mediated protein transfer to high-density lipoprotein (HDL). (**A**) Immunoblotted cellular (c) and medium (m) fractions of ShhN-expressing nt ctrl and Disp$^{-/-}$ cells. Shown is unspecific ShhN solubilization into serum-free media (upper blot, labeled [1]) or into serum-free Dulbecco's Modified Eagle's Medium (DMEM) supplemented with HDL (lower blot, labeled [2]). (**A'**) ShhN [1] expressed under serum-free conditions is solubilized in a monomeric state. (**A''**) ShhN [2] expressed in the presence of HDL remained monomeric (i.e., not HDL-associated). (**B**) Release of cholesterylated $^{C25S}$Shh into serum-free medium is very low (upper blot, labeled [3]), but increases in a Disp-dependent manner in the presence of HDL (lower blot, labeled [4]). Asterisks denote $^{C25S}$Shh solubilized independently of Disp function. (**B'**) Most $^{C25S}$Shh [3] in serum-free media is monomeric. (**B''**) $^{C25S}$Shh [4] expressed in the presence of HDL increases in molecular weight to match the molecular weight range of HDL (orange line, dotted brown line). (**C**) Size-exclusion chromatography (SEC) of $^{C25S}$Shh solubilized from Disp-expressing cells (solid green line) or from Disp$^{-/-}$ cells (dotted line) shows Disp-independent physical desorption and unregulated HDL association of the monolipidated protein. (**D**) SEC of cholesteroylated mCherry solubilized from nt ctrl cells (solid lines) or from Disp$^{-/-}$ cells (dotted lines) under the same conditions. Dashed lines indicate proteins solubilized under serum-free conditions. Note that most of the mCherry associates with HDL in a Disp-mediated manner. (**E**) $^{C25S}$Shh (green line) dissociates from HDL in 50% ethanol (bright green line) or in 0.1% Triton X-100 (bright green dashed line). The disassembly of HDL (orange line) under the same conditions is confirmed by the size shift of ApoA1 toward the monomeric 32 kDa protein (light orange line).

The online version of this article includes the following source data and figure supplement(s) for figure 7:

**Source data 1.** Raw data of *Figure 7*.

**Figure supplement 1.** Loading controls.

**Figure supplement 1—source data 1.** Uncropped western blots of *Figure 7—figure supplement 1*.

and its assembly to sizes similar to those of HDL (*Figure 7B and B"*, labeled [4]; see also *Figure 7—figure supplement 1C and D*). In support of the importance of Shh cholesterylation for its transfer to HDL, the C-termini of soluble $^{C25S}$Shh remained lipidated (*Figure 7—figure supplement 1E and F*). These results suggest that the C-terminal cholesterol moiety is required for Disp-mediated Shh transfer and HDL association. However, we also found that a fraction of monolipidated $^{C25S}$Shh was solubilized in a Disp-independent manner (*Figure 7B*, asterisks; *Figure 7—figure supplement 1F*, *Figure 7C*; the green dotted line indicates the distribution of $^{C25S}$Shh sizes when released from Disp$^{-/-}$ cells). We explain this finding by the unspecific release of $^{C25S}$Shh in the presence of HDL, consistent with increased desorption of large proteins – such as Shh – when attached only to a single membrane anchor (*Peters et al., 2004*). Once desorbed, lipidated proteins can either reintegrate into the plasma membrane or associate with lipophilic soluble competitors in a spontaneous manner, as shown in *Figure 7—figure supplement 1G*. This demonstrates that another important role of fully conserved

dual N- and C-terminal Hh lipidation is to reduce non-enzymatic desorption of morphogens from the plasma membrane at high LPP concentrations in the surrounding fluid.

Is the cholesteroylated peptide sufficient for Disp-mediated transfer to HDL or is the globular Shh ectodomain involved in the transfer (*Wang et al., 2021*; *Cannac et al., 2020*)? To answer this question, we replaced most of the C25SShh ectodomain with mCherry flanked N-terminally by the Shh secretion signal and C-terminally by the Shh cholesterol transferase/autoprocessing domain. This strategy resulted in the secretion of cholesteroylated mCherry to the cell surface and its association with the outer leaflet of the plasma membrane (*Figure 7—figure supplement 1H*). As previously observed for Shh (*Figure 5A"*), mCherry remained monomeric or formed small aggregates when released into serum-depleted media (*Figure 7D*, dashed violet line), and HDL shifted mCherry elution to fractions that also contained ApoA1 (solid violet/orange lines). This result supports that HDL serves as a soluble acceptor for C-terminally cholesterylated peptides. SEC also revealed that loss of Disp function decreased the relative amount of HDL-associated large clusters (*Figure 7D*, solid violet line) and increased the relative amount of solubilized monomeric protein (*Figure 7D*, dotted violet line). Thus, cholesterylated peptides represent the critical determinant for Disp-mediated Shh transport to HDL. In contrast, Shh ectodomain interactions with Disp are not essential for the transfer. We supported the physical interactions of C25SShh with HDL by the addition of EtOH or Triton X-100, both of which dissociate soluble high MW aggregates into monomers or small assemblies (*Figure 7E*, light green lines indicate C25AShh after treatment and the light orange line ApoA1 after treatment).

## Discussion
### Dual Shh lipidation and Disp-regulated Shh solubilization are interdependent

In the past, it has been well established that Hh solubilization and signaling depend on Disp co-expression in source cells (*Burke et al., 1999*; *Ma et al., 2002*; *Amanai and Jiang, 2001*). However, the

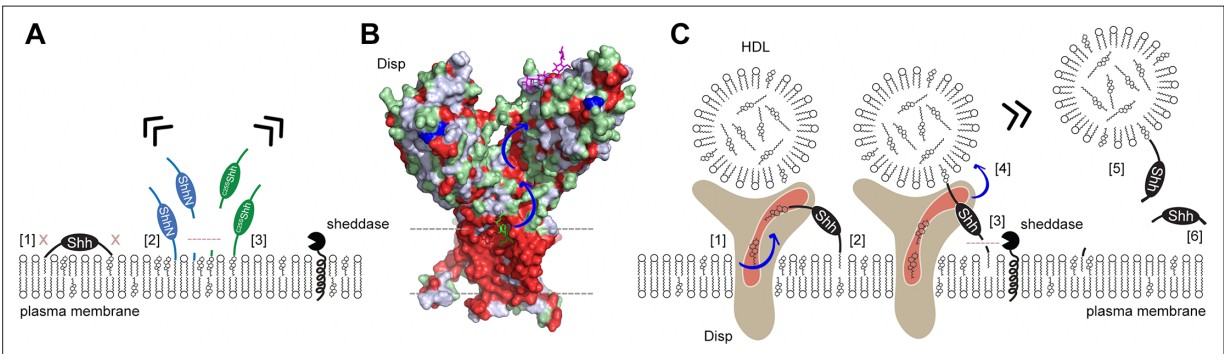

**Figure 8.** Model of two-way Disp-mediated Shh solubilization. (**A**) Dual lipidation protects Shh from unregulated cell surface shedding by tight plasma membrane association of both lipids (blocked shedding is indicated by an x in [1]). In contrast, monolipidated ShhN [2] and C25SShh [3] are prone to unregulated membrane proximal shedding (indicated by the dashed line) or non-enzymatic desorption. (**B**) Depiction of the surface hydrophobicity of Disp (pdb:7RPH; *Wang et al., 2021*) suggests an extended hydrophobic surface channel (hydrophobic residues are shown in red) that may function as a 'slide' for lipophiles extending from the plasma membrane (dashed lines) to a cavity of the second extracellular domain (blue arrows). A sterol lifted upward (green stick representation) at the starting point of the hydrophobic track may represent an intermediate state of sterol extraction from the membrane, and a lipidic group modeled as the amphiphilic detergent lauryl maltose neopentyl glycol (violet stick structure) may represent the end point of the transfer (*Wang et al., 2021*) prior to delivery to high-density lipoprotein (HDL). (**C**) We propose two sequences of Shh transfer events. In the first event [1], plasma membrane sterol is transferred through the hydrophobic Disp surface channel to HDL acceptors. This process is similar to the established reverse cholesterol transport. In the second event, if present, C-terminal cholesterol moieties of Shh can also be transferred [2]. This partial Shh extraction exposes the N-terminal cleavage site [3] and makes it susceptible to proteolytic processing (similar to ShhN as shown in **A**). N-terminal Shh shedding can then release the protein from the plasma membrane [4] to complete the transfer [5]. In addition to, or competing with this process, cholesterol depletion of the plasma membrane (representing the first event, [1]) may indirectly trigger shedding of both terminal Shh peptides and the solubilization of monomeric proteins [6], possibly as a consequence of the disruption of lipid rafts. See 'Discussion' for details.

The online version of this article includes the following source data and figure supplement(s) for figure 8:

**Figure supplement 1.** In vivo support for N-terminal shedding during Disp-mediated Hh export.

**Figure supplement 1—source data 1.** Statistical raw data of *Figure 8—figure supplement 1*.

additional involvement of many cell surface and soluble cofactors has complicated efforts to elucidate the exact nature of Disp-mediated Hh solubilization. To address this problem, we transfected Disp-deficient and Disp-expressing source cells (*Ehring et al., 2022*) with Shh and systematically modified Shh solubilization by a number of extracellular cofactors. Cell surface-associated Shh upstream of Disp function and soluble Shh downstream of Disp function were then analyzed together with their cofactors by using unbiased biochemical tools. The first important result of this approach is that both Shh lipids act together to prevent uncontrolled Shh desorption or shedding from producing cells. In contrast, engineered $^{C25S/C25A}$Shh, which lacks N-palmitate, and ShhN, which lacks C-cholesterol, are constitutively solubilized by either non-enzymatic desorption or unregulated shedding (*Figure 8A*). The second important finding is that Disp and Scube2 specifically and synergistically increase the shedding of the dual-lipidated cellular protein. These two observations suggest that cell surface shedding represents a 'ground state' from which dual-lipidated Shh is protected by tight plasma membrane association of both lipids – but only until Disp and Scube2 render Shh susceptible to proteolytic processing of the terminal lipidated peptides. This concept is important in reinterpreting the observation that dual lipidation is essential for unimpaired Hh biofunction in vivo (*Gallet et al., 2006*; *Porter et al., 1996a*; *Lewis et al., 2001*; *Huang et al., 2007*; *Lee et al., 2001*). The current interpretation of these observations is that both Hh lipids contribute directly to Ptch1 receptor binding and maximal signaling. Our results support an alternative mechanism acting upstream of Ptch1: we suggest that the selective artificial prevention of N- or C-terminal lipidation during Hh/Shh biosynthesis in vivo (as achieved by the deletion of the N-terminal palmitate acceptor cysteine or of Hhat, or by the prevention of C-cholesteroylation; *Gallet et al., 2006*; *Porter et al., 1996a*; *Lewis et al., 2001*; *Huang et al., 2007*; *Lee et al., 2001*) may have converted spatiotemporally controlled Hh/Shh solubilization into unregulated 'leakage' of monolipidated precursors from producing cells and tissues (*Figure 1*). The 'unavailability' of dual-lipidated Shh for Disp-controlled spatiotemporal signaling, or the desensitization of target cells to ongoing Shh exposure (*Dessaud et al., 2007*), may then have caused the observed Hh loss-of-function phenotypes (*Gallet et al., 2006*; *Porter et al., 1996a*; *Lewis et al., 2001*; *Huang et al., 2007*; *Lee et al., 2001*; *Ducuing et al., 2013*). Therefore, our release model does not contradict the established essential role of unimpaired dual Shh lipidation during biosynthesis for the regulation of Hh activity in vivo, but provides an alternative explanation for it. Published support for our model of Disp- and Scube2-regulated Hh shedding and release on HDL includes the following: Disp is required only in Hh-producing cells and not in receiving cells (*Tian et al., 2005*); Hh signaling defects in Disp-deficient model organisms are caused by a defect in the deployment of cell surface-associated Shh (*Kawakami et al., 2002*; *Ma et al., 2002*; *Burke et al., 1999*) and full Disp activity requires Scube2 (*Tukachinsky et al., 2012*). Further support comes from the in vivo observation that artificially depalmitoylated Hh variants impair development to varying degrees, but also show increased signaling in some tissues and induce ectopic signaling (*Lee et al., 2001*; *Li et al., 2006*). The latter observation is difficult to reconcile with the proposed essential signaling functions of the Shh palmitate to Ptch1, but is compatible with our model of spatiotemporally perturbed release of monolipidated proteins.

## Disp mediates Shh solubilization through two functionally related modes of transfer

The third important finding of our study is that the absence of all serum traces reduces or even abolishes Disp/Scube2-mediated Shh solubilization. This indicates that Disp and Scube2 are not sufficient to solubilize dual-lipidated Shh and that a serum factor is also required. We have previously shown that Disp knockout in cells impairs $^{3}$Hcholesterol efflux and increases membrane cholesterol levels, which in turn indirectly inhibits Shh shedding (*Ehring et al., 2022*). We also showed that HDL serves as a soluble sink for Disp-exported cholesterol and that Shh shedding from Disp$^{-/-}$ cells is restored by Disp and the established cholesterol transporter Ptch1 (*Ehring et al., 2022*). Consistent with these published observations, both Disp and Ptch1 contain sterol-sensing domains that are likely involved in cholesterol transport or its regulation (*Figure 8B*). Taken together, these published findings suggest that the required serum factor is HDL.

In this article, we describe that HDL also accepts cholesteroylated Shh from Disp (*Figure 8C*). This finding is supported by the presence of an extended hydrophobic surface channel in Disp that may function as an open 'slide' for larger lipophiles (*Figure 8B*, blue arrows). This slide may be powered by

a transmembrane Na$^+$ flux (**Wang et al., 2021**), similar to the H$^+$ flux that drives the related prokaryotic resistance-nodulation-division transporter export of small molecules (**Nikaido and Takatsuka, 2009**). The Disp exporter function of cholesteroylated proteins to HDL is further supported by the published concept that the fly LPP lipophorin carries cholesteroylated Hh in vivo (**Eugster et al., 2007**; **Palm et al., 2013**; **Panáková et al., 2005**). We extend this published concept by showing that the most C-terminal cholesteroylated Shh peptide is sufficient for direct Disp-mediated protein export to HDL because mCherry linked to this peptide is also transferred to HDL in a mostly Disp-dependent manner. This is consistent with a previous report showing that C-cholesterol is necessary and sufficient for Disp-mediated protein export (**Tukachinsky et al., 2012**). Due to their small size of 5–10 nm, HDLs are not only abundant in the circulation, but are also present in interstitial fluids, both in the adult and during development (**Palm et al., 2013**). In contrast, larger LPPs such as LDLs and VLDLs are limited in their distribution by vascular impermeability (**Lundberg et al., 2013**; **Randolph and Miller, 2014**). The ratio of HDL to LDL in interstitial fluids has been estimated to average 50:1 (**Randolph and Miller, 2014**). These properties make HDL well suited not only for reverse cholesterol transport, but also for the transport of cholesteroylated Hh/Shh cargo to its receptor Ptch1, which also functions as an LPP receptor (**Callejo et al., 2008**). This would lead to the interesting concept that 'Hh-free' LPPs and 'Hh-loaded' LPPs may compete for Ptch1 receptor binding, the former by promoting Ptch1-mediated cholesterol export to suppress signaling and the latter by terminating cholesterol efflux to trigger Hh signaling (**Zhang et al., 2018**). Indeed, this elegant mechanism of Ptch1 activity regulation has been previously demonstrated both in vitro and in vivo (**Palm et al., 2013**).

## Shedding of the N-terminus during Disp-mediated export of Hh in vivo

Our in vitro-derived concept of required N-terminal peptide shedding is supported by the in vivo finding that site-directed mutagenesis of the sheddase target site completely abolishes transgene function in the developing *Drosophila* wing and eye (see **Figure 8—figure supplement 1A and B** for a detailed description of repeated and combined experiments from published work; **Kastl et al., 2018**; **Schürmann et al., 2018**). Another striking observation was that the same mutant proteins suppress endogenous Hh biofunction in a dominant-negative manner (**Kastl et al., 2018**; **Schürmann et al., 2018**). However, the same site-directed mutagenesis approach at the Hh C-terminus only slightly affects the biofunction of the transgene and does not suppress endogenous Hh biofunction in vivo. Both observations can now be explained by the fact that mutant Hh transgenes readily associate their C-terminal peptides with the extended Disp 'slide' for lipophiles, with the N-mutant protein unable to complete the transfer due to blocked N-terminal shedding and continued plasma membrane association of the palmitoylated Hh N-terminus (preventing steps [3] and [4] due to transfer arrest at step [2] in **Figure 8C**). As a consequence, the resulting Disp bottleneck would slow down endogenous Hh release, explaining the observed dominant-negative developmental defects (**Figure 8—figure supplement 1C and D**; **Schürmann et al., 2018**; **Kastl et al., 2018**). This sequence of events is supported by reversed dominant-negative defects of the N-mutant protein upon additional removal of the palmitate membrane anchor (**Schürmann et al., 2018**; **Kastl et al., 2018**). Taken together, our results suggest that most Hh solubilization in vivo does not require C-terminal Hh shedding, but rather direct cholesterylated Hh transfer from Disp to LPPs. During this process, only the palmitoylated N-terminus is shed to complete the transfer (**Figure 8C**, **Figure 8—figure supplement 1C**). Our finding that cholesterylated C-terminal peptides are sufficient for Disp-mediated transfer to HDL is supported by the in vivo observation that transgenes with cholesterylated 27 kDa green fluorescent protein tags downstream of the 19 kDa Hh signaling domain are bioactive in flies (**Chen et al., 2017**) and mice (**Chamberlain et al., 2008**). Functional Disp specificity in vivo, however, can be elegantly explained by the fact that Hh is the only known metazoan protein with covalently attached cholesterol, and therefore the only substrate. Finally, our finding that palmitoylated Shh N-termini are not extracted and translocated by Disp is supported by the in vivo observation that transgenic expression of ShhN – the variant protein that is N-palmitoylated but lacks C-cholesterol – rescues many early Hh-related embryonic lethal defects in Disp$^{-/-}$ mutant mice (**Tian et al., 2005**).

# Materials and methods

**Key resources table**

| Reagent type (species) or resource | Designation | Source or reference | Identifiers | Additional information |
|---|---|---|---|---|
| Gene, fruit fly (*Drosophila melanogaster*) | Hedgehog; Hh | PMID:8252628 | NCBI ID:NM_001038976.1 | |
| Gene, fruit fly (*D. melanogaster*) | En-Gal4e16E (En>) | FlyBase; *Öztürk-Çolak et al., 2024* | FlyBase ID FBrf0098595 | *P(en2.4-GAL4)e16E* |
| Gene, fruit fly (*D. melanogaster*) | GMR-Gal4 (GMR>) | Bloomington *Drosophila* Stock Center (Indiana University) | Bloomington stock #45433 | *P(y[+t7.7]w[+mC]=GMR17G12-GAL4)attP2* |
| Gene, fruit fly (*D. melanogaster*) | UAS-hh, UAS-$^{HA}$hh, UAS-hh$^{HA}$ | Grobe Lab *Schürmann et al., 2018*; *Manikowski et al., 2020* | | Hh or Hh variant expression under UAS-control |
| Gene, fruit fly (*D. melanogaster*) | Low Hh expression in eye disc | Grobe Lab; *Kastl et al., 2018* | *w$^-$;+/+; hh$^{bar3}$/hh$^{AC}$* | Lacks Hh expression specifically in the eye disc |
| Cell line, human (*Homo sapiens*) | Bosc23 | Grobe Lab, provided by Dr. D. Robbins; *Zeng et al., 2001* | PMID:11395778 | HEK293 derivative |
| Cell line, human (*H. sapiens*) | Disp1 knockout cells (Disp$^{-/-}$) | Grobe Lab; *Ehring et al., 2022* | Bosc23 clone #17.10 | Lacks Disp expression |
| Cell line, human (*H. sapiens*) | nt ctrl cells | Grobe Lab; *Ehring et al., 2022* | Bosc23 clone #10.5 | Disp-expressing control cells |
| Cell line, human (*H. sapiens*) | Panc | American Type Culture Collection (ATCC) | ATCC CRL-2553 RRID:CVCL_0480 | Shh-expressing cancer cell line (pancreatic duct) |
| Cell line, murine (*Mus musculus*) | C17.2 | Merck | 07062902-1VL RRID:CVCL_4511 | Mouse multipotent neural progenitor or stem-like cells |
| Cell line, murine (*M. musculus*) | C3H10T1/2 | Grobe Lab (provided by Dr. Andrea Hoffmann, GBF Braunschweig, Germany) | | Multipotent Shh reporter cell line |
| Cell line, murine (*M. musculus*) | NIH3T3 | Leibniz-Institut DSMZ-Deutsche Sammlung von Mikroorganismen und Zellkulturen | DSMZ ACC 59 | Mouse fibroblast Shh reporter cell line |
| Transfected construct, murine (*M. musculus*) | pIRES | *Jakobs et al., 2014* | Clontech 631605 | Bicistronic expression vector |
| Biological sample, human (*H. sapiens*) | HDL | Merck Millipore, Burlington, NH | Millipore, LP3, MW 175,000-360,000 | |
| Biological sample, human (*H. sapiens*) | LDL | Merck Millipore, Burlington, NH | Millipore, LP2, MW 2,300,000 Da | |
| Antibody | α-Shh (rabbit monoclonal) | Cell Signaling, Danvers, MA | C9C5 RRID:AB_2188191 | Used to detect cellular and solubilized Shh (1:5000) |
| Antibody | α-GAPDH (rabbit polyclonal) | Cell Signaling, Danvers, MA | 14C10, #2118 RRID:AB_1903993 | Loading control (1:2000) |
| Antibody | α-β-actin HRP conjugated (mouse monoclonal) | Sigma-Aldrich, St. Louis, MO | A3854 RRID:AB_262011 | Loading control (1:10,000) |
| Antibody | α-FLAG (rabbit polyclonal) | Sigma-Aldrich, St. Louis, MO | F7425 RRID:AB_439687 | Used to detect FLAG-tagged Scube2 (1:5000) |
| Antibody | α-mCherry antibodies (rabbit polyclonal) | Thermo Fisher Scientific, Rockford, IL | PA5-34974 RRID:AB_2552323 | (1:2000) |

*Continued on next page*

*Continued*

| Reagent type (species) or resource | Designation | Source or reference | Identifiers | Additional information |
|---|---|---|---|---|
| Antibody | α-Shh antibody 5E1 (mouse monoclonal) | Developmental Studies Hybridoma Bank (DSHB), Iowa City, IA | DSHB 5E1 RRID:AB_528466 | Binds to Shh pseudoactive site that also binds Ptch1 (1:1000) |
| Antibody | α-ApoA1 (mouse monoclonal) | NovusBio, Wiesbaden, Germany | NB400-147 RRID:AB_10001123 | Detects HDL (integral protein) (1:1000) |
| Antibody | α-ApoE4 (mouse monoclonal) | Cell Signaling, Danvers, MA | (4E4) #2208 RRID:AB_2238543 | Detects HDL (mobile protein) (1:1000) |
| Recombinant DNA reagent | Shh (murine) | Grobe Lab | OriGene NM_009170 | Dual-lipidated Shh |
| Recombinant DNA reagent | ShhN/$^{C25A/S}$Shh | Grobe Lab | OriGene NM_009170 | Monolipidated Shh |
| Recombinant DNA reagent | Hhat | Grobe Lab | OriGene NM_018194 | |
| Recombinant DNA reagent | V5-tagged Disp$^{tg}$ | Stacey Ogden (St. Jude Children's Research Hospital, Memphis, USA) | | |
| Recombinant DNA reagent | Ptch1$^{tg}$ and Ptch1$^{ΔL2}$ | *Ehring et al., 2022* | Addgene_#120889 | |
| Recombinant DNA reagent | pNH-NanoLuc | Addgene, Watertown, MA | #173075 | |
| Peptide, recombinant protein | HEK293-derived human Shh | R&D Systems, Minneapolis, MN | 8908-SH | Dual-lipidated Shh extracted from transfected HEK cells |
| Commercial assay or kit | Gibson assembly | New England Biolabs, Frankfurt, Germany | HiFi Assembly Kit, NEB # E5520S | |
| Commercial assay or kit | Mouse mesenchyme stem cell functional id kit | R&D Systems, Minneapolis, MN | R&D Systems, SC010 | |
| Chemical compound, drug | [$^3$H]-cholesterol | PerkinElmer, Waltham, MA | NET139250UC | |
| Software, algorithm | ImageJ | *Schroeder et al., 2021* | Version 1.54g | Immunoblot quantification |
| Software, algorithm | GraphPad Prism | GraphPad Software, Boston, MA , https://graphad.com | Version 10.2.3 (347) | |
| Other | Protein A beads | Sigma-Aldrich, St. Louis, MO | P1406 | ProteinA derived from *Staphylococcus aureus* coupled to agarose beads for IgG antibody binding and subsequent immunoprecipitation |

## Fly lines

The fly lines *En-Gal4e16E* (*En>*): *P(en2.4-GAL4)e16E*, FlyBaseID FBrf0098595 and *GMR-GAL4 (GMR>): GMR17G12 (GMR45433-GAL4): P(y[+t7.7]w[+mC]=GMR17G12-GAL4)attP2*, Bloomington stock #45433 (discontinued but available from our lab), were crossed with flies homozygous for *UAS-hh* (*Schürmann et al., 2018*) or variants thereof (previously published in *Manikowski et al., 2020*; *Kastl et al., 2018*; *Schürmann et al., 2018*). Shh cDNA cloned into pUAST-attP was first expressed in *Drosophila* S2 cells to confirm correct protein processing and secretion. Transgenic flies were generated by using the landing site *51C1* by BestGene. Driver lines were crossed with flies homozygous for *UAS-hh* or variants thereof and kept at 25°C unless otherwise noted. Cassette exchange was mediated by germ-line-specific phiC31 integrase (*Bateman et al., 2006*). *w;+/+;hh$^{bar3}$/hh$^{AC}$* flies served as negative controls; *white$^{1118}$* flies served as positive controls. Eye phenotypes were analyzed with a Nikon SMZ25 microscope.

## Cholesterol efflux assay

To conduct this assay, we followed a published protocol (*Low et al., 2012*). Briefly, Disp$^{-/-}$ cells and nt ctrl cells were seeded in 12-well plates at a final density of $0.2 \times 10^6$ cells per well in 0.9 mL DMEM containing 10% FCS and 100 µg/mL penicillin-streptomycin, and cells were incubated at 37°C, 5% $CO_2$. After 24 hr, the medium was changed for DMEM containing 10% FCS, 100 µg/mL penicillin-streptomycin, and 0.5 µCi [$^3$H]-cholesterol (PerkinElmer, Foster City, USA) per well. After 2 days, media containing the [$^3$H]-cholesterol were removed, the cells were gently washed, and serum-free media with 0.1% BSA was added. After 18 hr, cells were checked under the microscope for confluency and the medium exchanged for 250 µL serum-free medium or media containing 0.05, 5, and 10% FCS. After 3 hr, cells and media were harvested and transferred into scintillation vials, [$^3$H] signals were counted, and the amount of released [$^3$H]-cholesterol was expressed as the proportion of solubilized [$^3$H]-cholesterol detected in the media (minus the blank efflux) divided by the cellular [$^3$H]-cholesterol amounts after normalization for protein content.

## Cell lines

The generation and validation of Disp1 knockout cells (Disp$^{-/-}$) and nt ctrl cells were previously described (*Ehring et al., 2022*). Disp$^{-/-}$, nt ctrl, C3H10T1/2, and NIH3T3 reporter cells were maintained in DMEM supplemented with 10% FCS and 100 µg/mL penicillin-streptomycin. All cell lines were tested negative for mycoplasma.

## Cloning of recombinant proteins

Shh expression constructs were generated from murine cDNA (NM_009170: nucleotides 1–1314, corresponding to amino acids 1–438; and ShhN: nucleotides 1–594, corresponding to amino acids 1–198) and human Hhat cDNA (NM_018194). Both cDNAs were cloned into pIRES (Clontech) for their coupled expression from bicistronic mRNA to achieve near-quantitative Shh palmitoylation (*Jakobs et al., 2014*). ShhN (nucleotides 1–594, corresponding to amino acids 1–198) and Hhat were also cloned into pIRES. $^{C25S}$Shh was generated by site-directed mutagenesis (Stratagene). Unlipidated $^{C25S}$ShhN cDNA and non-palmitoylated $^{C25S}$Shh cDNA (amino acids 1–438) were inserted into pcDNA3.1 (Invitrogen). Primer sequences can be provided upon request. Human Scube2 constructs were a kind gift from Ruey-Bing Yang (Academia Sinica, Taiwan). Murine V5-tagged Disp$^{tg}$ was a kind gift from Stacey Ogden (St. Jude Children's Research Hospital, Memphis, USA). Murine Ptch1$^{tg}$ and Ptch1$^{\Delta L2}$ were generated from Ptch1 Full Length (pcDNA-h-mmPtch1-FL, Addgene #120889). Ptch1$^{\Delta L2}$ was generated by deletion of the second extracellular loop (L2) between transmembrane domains 7 and 8 (amino acids 794–997). Primer sequences can be provided upon request. For Shh-NanoLuc, NanoLuc (pNH-NanoLuc, Plasmid #173075, Addgene), flanked by one glycine residue on both sides, was inserted into murine Shh between amino acids 92N and 93T (corresponding to N91 and T92 in human Shh) by using Gibson assembly (HiFi Assembly Kit, NEB). Where indicated, dual-lipidated, HEK293-derived human Shh (R&D Systems, 8908-SH) served as a bioactivity control and to quantify Bosc23-expressed, TCA-precipitated proteins on the same blot.

## Protein detection

Bosc23 cells, nt ctrl cells, or Disp$^{-/-}$ cells were seeded into six-well plates and transfected with 1 µg Shh constructs together with 0.5 µg Scube2 or empty cDNA3.1 using Polyfect (QIAGEN). Where indicated, 0.5 µg Disp or Ptch1 encoding constructs were co-transfected. Cells were grown for 36 hr – 2 days at 37°C with 5% $CO_2$ in DMEM containing 10% FCS and penicillin-streptomycin (100 µg/mL). Where indicated, 50 µM peptidyl-CMK (Millipore 344930), an inhibitor of furin activity in DMSO, or DMSO alone was added to the media. Serum-containing media were aspirated and serum-free DMEM added for 6 hr, harvested, and centrifuged at 300 × $g$ for 10 min to remove debris. Supernatants were incubated with 10% trichloroacetic acid (TCA) for 30 min on ice, followed by centrifugation at 13,000 × $g$ for 20 min to precipitate the proteins. Proteins solubilized into serum-containing media were pulled down by using heparin-sepharose beads (Sigma). Cell lysates and corresponding supernatants were analyzed on the same reducing SDS polyacrylamide gel and detected by western blot analysis by using rabbit-α-Shh antibodies (Cell Signaling C9C5), rabbit-α-GAPDH antibodies (Cell Signaling, GAPDH 14C10, #2118), or anti-β-actin antibodies (Sigma-Aldrich, A3854), followed by incubation with horseradish peroxidase-conjugated secondary antibodies. Flag-tagged Scube2 was detected using

polyclonal α-FLAG antibodies (Sigma, St. Louis, USA). GAPDH, β-actin (for cell lysates), or PonceauS (for media) served as a loading control. Note that the amounts of immunoblotted soluble and cellular Shh do not correlate inversely. This is because medium lanes represent all TCA-precipitated proteins or all proteins captured by heparin-pulldown in medium, while cells were directly lysed in SDS buffer and only a small fraction (about 5%) were applied to the gel. As a consequence, a 100% increase in Shh solubilization will correlate to only 5% reduction in the amount of cell-surface-associated Shh. Shh release was quantified using ImageJ and calculated as the ratio of total, unprocessed (top band) or processed (truncated, bottom band) soluble Shh relative to the corresponding cellular Shh material and multiplied by 100 to express as a percentage. This protocol was varied in three ways: for *serum-free* release, cells cultured in DMEM + 10% FCS were carefully washed three times with serum-free DMEM before serum-free media were added for 6 hr of protein release. For Shh release into *serum-depleted* medium, cells were not washed before the serum-free DMEM was added. For release into *serum-containing* media, DMEM containing the indicated amounts of serum was added for 6 hr. For mCherry visualization at the surface of Bosc23 cells, cells were incubated with polyclonal anti-mCherry antibodies (Invitrogen PA5-34974) under non-permeabilizing conditions, and mCherry was visualized by secondary anti-rabbit IgG (Dianova) using a Zeiss LSM700 confocal microscope.

## Shh release in the presence of HDL or LDL

nt ctrl or Disp$^{-/-}$ cells were transfected with pIRES for coupled Shh and Hhat expression, together with Scube2 cDNA as described earlier. Two days after transfection, cells were washed twice with serum-free DMEM and additionally incubated for 1 hr in serum-free DMEM. This extensive washing was intended to quantitatively remove serum LPPs. Serum-free DMEM was then discarded and cells were incubated in serum-free DMEM containing 80 μg/mL human HDL (Millipore, LP3, MW 175,000–360,000) or LDL (Millipore, LP2, MW 2,300,000 Da) at 80 μg/mL or at similar molarity (630 μg/mL) for 6 hr. Increased Shh release was observed for HDL concentrations ranging from 40 μg/mL to 120 μg/mL; higher HDL concentrations were not tested. For cell debris removal, supernatants were centrifuged for 10 min at 300 × *g*. For subsequent Shh purification, supernatants were TCA precipitated or incubated with 5 μg/mL anti-Shh antibody DSHB 5E1 for 2 hr at 4°C, followed by the addition of 5 mg protein A beads (Sigma, P1406) in PBS and incubated at 4°C overnight. Immunoprecipitates were collected by centrifugation at 300 × *g* for 5 min and subjected to reducing SDS-PAGE followed by immunoblot analysis. Shh release was quantified by first determining the ratios of soluble Shh signals detected in 5E1-Protein A pulldown samples relative to cellular Shh signals. Shh release from nt ctrl and Disp$^{-/-}$ cells was next compared, with nt ctrl release set to 100%. Where indicated, $^{C25S}$Shh/HDL assemblies were dissolved in 50% ethanol or 0.1% Triton X-100 for 2 min before SEC analysis. Immunoblotted HDL was identified by using antibodies directed against apolipoprotein (Apo)A1 (NB400-147, NovusBio) and mobile ApoE4 (4E4 Cell Signaling), the latter engaging in HDL size expansion (*Sacks and Jensen, 2018*).

## SEC chromatography

Shh size distribution in the presence or absence of soluble carriers was confirmed by SEC analysis with a Superdex200 10/300 GL column (GE Healthcare, Chalfont St. Giles, UK) equilibrated with PBS at 4°C fast protein liquid chromatography (Äkta Protein Purifier, GE Healthcare). Eluted fractions were TCA-precipitated, resolved by 15% SDS-PAGE, and immunoblotted. Signals were quantified with ImageJ. When indicated, eluted fractions were split and one half used for Shh activity determination.

## Density gradient (isopycnic) centrifugation

For OptiPrep (a 60% [w/v] solution of iodixanol in water, density = 1.32 g/mL) gradients, Shh and $^{C25A}$Shh were solubilized in the presence of 40 μg/mL human HDL overnight, the medium centrifuged at 10,000 rpm for 10 min to remove cellular debris, and adjusted to 17.6% OptiPrep/iodixanol. Solutions of 15 and 23% OptiPrep were layered on top or below the sample and centrifuged at 4°C for 16 hr at 120,000 × *g* in a SW 28 Ti swinging bucket rotor (Beckman).

## Shh bioactivity assay

SEC fractions from Shh expressed into serum-containing media or DMEM supplemented with HDL were sterile filtered, FCS was added to the fractions at 10% and mixed 1:1 with DMEM supplemented

with 10% FCS and 100 µg/mL antibiotics, and the mixture was added to C3H10 T1/2 cells. Cells were harvested 6 days after osteoblast differentiation was induced and lysed in 1% Triton X-100 in PBS, and osteoblast-specific alkaline phosphatase activity was measured at 405 nm by using 120 mM $p$-nitrophenolphosphate (Sigma) in 0.1 M Tris buffer (pH 8.5). Values measured in mock-treated C3H10 T1/2 cells served as negative controls and were subtracted from the measured values.

### Quantitative PCR (qPCR)

C3H10T1/2 or NIH3T3 cells were stimulated with recombinant Shh in triplicate and media were exchanged every 3–4 days. TRIzol reagent (Invitrogen) was used for RNA extraction from C3H10T1/2 cells 1 day after Shh stimulation or 5 days after Shh stimulation. A first-strand DNA synthesis kit and random primers (Thermo, Schwerte, Germany) were used for cDNA synthesis before performing a control PCR with murine β-actin primers. Amplification with Rotor-Gene SYBR-Green on a Bio-Rad CFX 384 machine was conducted in triplicate according to the manufacturer's protocol by using the primer sequences listed in *Supplementary file 3*. Cq values of technical triplicates were averaged, the difference to β-actin mRNA levels calculated by using the ΔΔCt method, and the results expressed as log2-fold change if compared with the internal control of C3H10T1/2 or NIH3T3 cells stimulated with mock-transfected media.

### Reverse-phase high-performance liquid chromatography (RP-HPLC)

Bosc23 cells were transfected with expression plasmids for dual-lipidated Shh, unlipidated $^{C25A}$ShhN control protein, cholesteroylated (non-palmitoylated) $^{C25A}$Shh, and palmitoylated ShhN. Two days after transfection, cells were lysed in radioimmunoprecipitation assay buffer containing complete protease inhibitor cocktail (Roche, Basel, Switzerland) on ice and ultracentrifuged, and the soluble whole-cell extract was acetone precipitated. Protein precipitates were resuspended in 35 µL of (1,1,1,3,3,3) hexafluoro-2-propanol and solubilized with 70 µL of 70% formic acid, followed by sonication. RP-HPLC was performed on a C4-300 column (Tosoh, Tokyo, Japan) and an Äkta Basic P900 Protein Purifier. To elute the samples, we used a 0–70% acetonitrile/water gradient with 0.1% trifluoroacetic acid at room temperature for 30 min. Eluted samples were vacuum dried, resolubilized in reducing sample buffer, and analyzed by SDS-PAGE and immunoblotting. Proteins expressed into the media were analyzed in the same way. Signals were quantified with ImageJ and normalized to the highest protein amount detected in each run. Where indicated, Disp$^{-/-}$ cells were incubated for 24 hr instead of the standard 6 hr for protein expression before media harvest and TCA precipitation to compensate for the very low Shh release rate from these cells.

### Bioanalytical and statistical analysis

All statistical analyses were performed in GraphPad Prism. Applied statistical tests, post hoc tests, and number of independently performed experiments are stated in the figure legends. A p-value of <0.05 was considered statistically significant. *p<0.05, **p<0.01, ***p<0.001, and ****p<0.0001 in all assays. Error bars represent the standard deviations of the means. Standard deviations as shown for Shh protein expression, and release from nt ctrl cells on western blots represents their variations from the average value (set to 100%) detected on the same blot.

## Acknowledgements

The excellent technical work of Sabine Kupich and Reiner Schulz is gratefully acknowledged. We thank Dr. S Ogden (St Jude Children's Research Hospital, Memphis, TN, USA) for Disp cDNA. We also acknowledge the pioneering work of Suzanne Eaton on lipoprotein-mediated Hh transport in the fly. This work was funded by German Research Foundation (DFG) grants SFB1348A08, GR1748/7-1, and GR1748/9-1, and MedK 20-0012 support of the Medical Faculty of the University of Münster (to SFE). We acknowledge support by the Open Access Publication Fund of the University of Münster.

## Additional information

### Funding

| Funder | Grant reference number | Author |
|---|---|---|
| Deutsche Forschungsgemeinschaft | SFB1348A08 | Kay Grobe |
| Deutsche Forschungsgemeinschaft | GR1748/7-1 | Kay Grobe |
| Deutsche Forschungsgemeinschaft | GR1748/9-1 | Kay Grobe |
| MedK | 20-0012 | Sophia Friederike Ehlers |

The funders had no role in study design, data collection and interpretation, or the decision to submit the work for publication.

### Author contributions

Kristina Ehring, Formal analysis, Validation, Investigation; Sophia Friederike Ehlers, Formal analysis, Investigation, Visualization; Jurij Froese, Fabian Gude, Janna Puschmann, Investigation; Kay Grobe, Conceptualization, Resources, Supervision, Funding acquisition, Visualization, Writing – original draft, Project administration, Writing – review and editing

### Author ORCIDs

Kay Grobe (ID) https://orcid.org/0000-0002-8385-5877

Reviewer #1 (Public Review): https://doi.org/10.7554/eLife.86920.4.sa1
Reviewer #2 (Public Review): https://doi.org/10.7554/eLife.86920.4.sa2
Author response https://doi.org/10.7554/eLife.86920.4.sa3

---

## Additional files

### Supplementary files

• Supplementary file 1. Raw data used for the figures in this study.
• Supplementary file 2. qPCR target genes analyzed in this work.
• Supplementary file 3. Information regarding amplicons and primers.
• MDAR checklist

### Data availability

All data are available in the main text or the supplementary materials and accompanying source data files. Materials can be made available upon request.

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
