## [Editor Report · eLife assessment]

This **useful** article presents an analysis of different factors that are required for release of the lipid-linked morphogen Shh from cellular membranes. The evidence is still **incomplete**, as experiments rely on over-expression of Shh in a single cell line and are sometimes of a correlative nature. The study, which otherwise confirms and extends previous findings, will be of interest to developmental biologists who work on Hedgehog signaling.

---

## [Referee Report · Reviewer #1 (Public Review)]

This manuscript presents a model in which combined action of the transporter-like protein DISP and the sheddases ADAM10/17 promote shedding of a mono-cholesteroylated Sonic Hedgehog (SHH) species following cleavage of palmitate from the dually lipidated precursor ligand. The authors propose that this leads to transfer of the cholesterol-modified SHH to HDL for solubilization. The minimal requirement for SHH release by this mechanism is proposed to be the covalently linked cholesterol modification because DISP could promote transfer of a cholesteroylated mCherry reporter protein to serum HDL. The authors used an in vitro system to demonstrate dependency on DISP/SCUBE2 for release of the cholesterol modified ligand. These results confirm previously published results from other groups (PMC3387659 and PMC3682496).

A strength of the work is the use of a bicistronic SHH-Hhat system to consistently generate dually-lipidated ligand to determine the quantity and lipidation status of SHH released into cell culture media.

Key shortcomings include the unusual normalization strategies used for many experiments and the lack of quantification/statistical analyses for several experiments. Due to these omissions, it is difficult to conclude that the data justify the conclusions. The significance of the data provided is overstated because many of the presented experiments confirm/support previously published work. The study provides a modest advance in understanding of the complex issue of SHH membrane extraction.

---

## [Referee Report · Reviewer #2 (Public Review)]

Ehring et al. analyze contributions of Dispatched, Scube2, serum lipoproteins and Sonic Hedgehog lipid modifications to the generation of different Shh release forms. Hedgehog proteins are anchored in cellular membranes by N-terminal palmitate and C-terminal cholesterol modifications, yet spread through tissues and are released into the circulation. How Hedgehog proteins can be released, and in which form, remains controversial. The authors systematically dissect contributions of several previously identified factors, and present evidence that Disp, Scube2 and lipoproteins concertedly act to release a novel Shh variant that is cholesterol-modified but not palmitoylated. The results provide new insights into the function of Disp and Scube2 in Hedgehog release. The findings concerning the function of lipoproteins and cholesterol in Hedgehog release are largely confirmatory (PMID 23554573, 20685986). However, in light of the multitude of competing models for Hedgehog release, the present study is a valuable contribution that provides further insights into the relevance of lipoproteins in this process.

A novel and surprising finding of the present study is the differential removal of Shh N- or C-terminal lipid anchors depending on the presence of HDL and/or Disp. In particular, the identification of a non-palmitoylated but cholesterol-modified Shh variant that associates with lipoproteins is potentially important. The authors use RP-HPLC and defined controls to assess the properties of processed Shh forms, but their precise molecular identity remains to be defined. A caveat is the strong reliance on over-expression of Shh in a single cell line. The authors detect Shh variants that are released independently of Disp and Scube2 in secretion assays, which however are excluded from interpretation as experimental artifacts. Thus, it would be important to demonstrate key findings in cells that secrete Shh endogenously.

---

## [Author Response]

The following is the authors’ response to the current reviews.

**Reviewer #1 (Public Review):**
Major shortcomings include the unusual normalization strategies used for many experiments and the lack of quantification/statistical analyses for several experiments. Because of these omissions, it is difficult to conclude that the data justify the conclusions. The significance ofthe data presented is overstated, as many of the experiments presented confirm/supportpreviously published work. The study provides a modest advance in the understanding of the complex issue of SHH membrane extraction.Major shortcomings include the unusual normalization strategies used for many experiments and the lack of quantification/statistical analysis for several experiments.

This statement is not any more correct for the revised manuscript: The normalization strategies used are clearly described in the manuscript and are not unusual. Each experiment is now statistically analyzed.

The significance of the data presented is overstated, as many of the experiments presented confirm/support previously published work.

As reviewer 2 correctly points out, there are many competing models for Hedgehog release. Our study does not support them all - the reviewer's statement is therefore misleading. In fact, our careful biochemical analysis of the mechanistics of Dispatched- mediated Shh export supports only two of them: The model of proteolytic processing of Shh lipid anchors (shedding) and the model of lipoprotein-mediated Shh transport. In contrast, our study does not support the predominant model of Dispatched-mediated extraction of dual-lipidated Shh and delivery to Scube2, which is currently thought to act as a soluble Shh chaperone. We also do not support Dispatched function in Shh endocytic recycling and cytoneme loading, or any of the other models such as exosome-mediated or micelle Shh transport.

**Reviewer #2 (Public Review):**
A novel and surprising finding of the present study is the differential removal of Shh N- or C- terminal lipid anchors depending on the presence of HDL and/or Disp. In particular, theidentification of a non-palmitoylated but cholesterol-modified Shh variant that associates with lipoproteins is potentially important. The authors use RP-HPLC and defined controls to assess the properties of processed forms of Shh, but their precise molecular identity remains to be defined. One caveat is the heavy reliance on overexpression of Shh in a single cell line.The authors detect Shh variants that are released independently of Disp and Scube2 in secretion assays, but these are excluded from interpretation as experimental artifacts. Therefore, it would be important to demonstrate key findings in cells that endogenously secrete Shh.

We would like to respond as follows:

The authors use RP-HPLC and defined controls to assess the properties of processed forms of Shh, but their precise molecular identity remains to be defined.

This statement refers to our original manuscript submission. We believe that the biochemical and functional data presented in the VOR clearly describe the molecular identity of solubilized Shh: it is monolipidated, lipoprotein-associated via its C-terminal cholesterol moiety, and highly biologically active in two established Shh bioassays.

One caveat is the heavy reliance on overexpression of Shh in a single cell line.

As stated by reviewer 1, the strength of our work is the use of a bicistronic SHH-Hhat system to consistently generate doubly lipidated ligand to determine the amount and lipidation status of SHH released into cell culture media. This unique system therefore eliminates the most concerning artifact of Shh overexpression -tha lack of quantitative N-acylation. Per this reviewers request, we have also added two other cell lines to our VOR that produce the same results (including Panc1 cells that endogenously produce Shh, Supplementary Figure 1).

The authors detect Shh variants that are released independently of Disp and Scube2 in secretion assays, but these are excluded from interpretation as experimental artifacts.

As the reviewer correctly points out, these variants are released independently of Disp and Scube2, both of which are known as essential release factors in vivo. These variants are therefore by definition experimental artifacts. The proteins that we have included in our analysis are the alternative forms that clearly depend on Dispatched and Scube2 for their release - as shown in the first figure in the manuscript, and in pretty much every other figure after that.

The following is the authors’ response to the previous reviews.

**Reviewer #1 (Public Review):**
Key shortcomings include the unusual normalization strategies used for many experiments and the lack of quantification/statistical analyses for several experiments.

In the updated version of the paper, we have addressed all of this reviewer's criticisms. Most importantly, we have performed several additional experiments to address the concern that unusual normalization strategies were used in our paper and that quantification and statistical analyses were lacking for several experiments. We have now analyzed the full set of release conditions for Shh and engineered proteins from Disp-expressing n.t. control cells and Disp-/- cells both in the presence and absence of Scube2 (Figure 1A'-D', Figure 2E added to the paper, Figure 3B'-D', Figure 5C and Figure 1 - figure supplement 2 K-M). Previously, we had only quantified protein release from n.t. controls and Disp-/- cells in the presence but not in the absence of Scube2 under serum-depleted conditions. Quantifications of serum-free protein release and Shh release under conditions ranging from 0.05% FCS to 10% FCS were completely missing from the earlier versions of the manuscript, but have now been added to our paper. In addition, we have reanalyzed all of the data sets in the above figures, as well as Figures 2C and S1B, to address the issue of "unusual normalization strategies": unlike previous assays in which the highest amount of protein detected in the media was set to 100% and all other proteins in that experiment were expressed relative to that value, we now directly compare the relative amounts of cellular and corresponding solubilized proteins as a method to quantify release without the need for data normalization (Figs. 1A'-D', 2C,E, 3B'-D', E, 5C, Figure 1 - figure supplement 1 B, Figure 1 - figure supplement 2 K-M).

We have also repeated the qPCR analyses in C3H10T1/2 cells and now show that the same Shh/C25AShh activities can be observed when using another Shh responsive cell line, NIH3T3 cells (Fig. 4B, 6B, FFigure 4 - figure supplement 1 B).

We would like to point out that if the criticism refers to the presentation of our RP-HPLC and SEC data, the normalization of the strongest eluted protein signal to 100% for all proteins tested is necessary to put their behavior in a clearer relationship. This is because only the relative positions of protein elution, and not their amounts, are important in these experiments.

The significance of the data provided is overstated because many of the presented experiments confirm/support previously published work.

To mitigate the first reviewer's comment that the significance of the data presented is overstated, we now clearly distinguish between our novel results and the known aspect of Hh release on lipoproteins throughout our paper. We now clearly describe what is new and important in our paper: First, contrary to the general perception in the field, Disp and Scube2 are not sufficient to solubilize Shh, casting doubt on the currently accepted model that Scube2 accepts dual-lipidated Shh from Disp and transports it to the receptor Ptch. Second, lipoproteins shift dual Shh processing to N-terminal peptide processing only to generate different soluble Hh forms with different activities (as shown in Figure 4C). Third, and again contrary to popular belief, this new release mode does not inactivate Shh, as we now show in two established cellular assays for Hh biofunction (Figures 4A-C, 5B'', 6B and Figure 4 - figure supplement 1 C-G). Fourth, and most importantly, we show that spatiotemporally controlled, Disp-, Scube2- and HDL-mediated Shh release absolutely requires dual lipidation of the membrane-associated Shh precursor prior to its release. This finding (as shown in Figures 1 and Figure 1 - figure supplement 2) changes the interpretation of previously published in vivo data that have long been interpreted as evidence for the requirement of dual Shh lipidation for full receptor binding and activation.

The study provides a modest advance in our understanding of the complex issue of Shh membrane extraction.

Although we agree that our results integrate our novel observations into previously established concepts of Hh release and trafficking, we also hope that our data cast well-founded doubt on the current view that the issue of Hh release and trafficking is largely resolved by the model of Disp-mediated Shh hand-over to Scube2 and then to Ptch, which requires interactions with both Shh lipids. Our data show that this is clearly not the case in the presence of lipoproteins. Thus, the significance of our data is that models of Shh lipid-regulated signaling to Ptch that were obtained using the dual-lipidated Shh precursor prior to its Disp- and Scube2-mediated conversion into a delipidated or monolipidated, HDL-associated soluble ligand are likely to describe a non-physiological interaction. Instead, our work describes a highly bioactive soluble ligand with only one lipid still attached, which has not been described before in the literature. The in vivo endpoint analyses presented in Figure 8 - figure supplement 1A and B suggest that this new protein variant is likely to play an important role during development.

**Reviewer #2 (Public Review):**
The precise molecular identity (of the released Shh) remains to be defined.

We would like to respond that the direct comparison of soluble proteins and their well-defined double-lipidated precursors side-by-side in the same experiment, as shown in our paper, determines all relevant molecular changes in the Shh release process. Most importantly, we show by SDS-PAGE and RP-HPLC that HDL restricts Shh processing to the N-terminus and that the absence of HDL results in double processing of Shh during its release. We also show by SEC that the C-terminus binds the Shh protein to HDL. In addition, the fly experiments confirm the requirement for N-terminal Hh processing, but not for processing of the C-terminal peptide, and suggest that the N-terminal Cardin-Weintraub sequence replaced by the functionally blocking tag represents the physiological cleavage site.

It would be important to demonstrate key findings in cells that secrete Shh endogenously.

We now confirm the key findings of our study in Panc1 cells that endogenously produce and secrete Shh: As shown in Figure 1 - figure supplement 1 D, we find that soluble proteins are processed but retain the C-cholesterol, which we now directly confirm by RP-HPLC (Figure 3 - figure supplement 2 F-H). The in vivo analyses shown in Figure 8 - figure supplement 1 A and B suggest that the key finding - that N-terminal but not C-terminal Hh shedding is required for release - can be supported, at least in the fly: here, Hh variants impaired in their ability to be processed N-terminally strongly repress the endogenous protein, whereas the same protein impaired in its ability to be processed C-terminally does not.

The authors detect Shh variants that are expressed independently of Disp and Scube2 in secretion assays, but are excluded from interpretation as experimental artifacts.

We agree with the reviewer's comment that the amounts of Shh released independently of Disp and Scube2 in secretion assays were initially not quantified and analyzed statistically to justify their proposed status as not physiologically relevant. We now show that these forms are indeed secretion artifacts. Fig. 3E and Figure 1 - figure supplement 2 K-M show quantification of the lower electrophoretic mobility protein fraction (i.e., the "top" band representing the double-lipidated soluble protein fraction). This fraction is released independently of Disp and Scube2.